



# Climatic history of the northeastern United States during the past 3000 years

Jennifer R. Marlon[1], Neil Pederson[2], Connor Nolan[3], Simon Goring[4], Bryan Shuman[5], Robert Booth[6], Patrick J. Bartlein[7], Melissa A. Berke[8], Michael Clifford[9], Edward Cook[10], Ann Dieffenbacher-Krall[11], Michael C. Dietze[12], Amy Hessl[13], J. Bradford Hubeny[14], Stephen T. Jackson[3,15], Jeremiah Marsicek[5], Jason McLachlan[16], Cary J. Mock[17], David J.P. Moore[18], Jonathan Nichols[19], Ann Robertson[1], Kevin Schaefer[20], Valerie Trouet[21], Charles Umbanhowar[22], John W. Williams[4], and Zicheng Yu[6]

[1]Yale School of Forestry and Environmental Studies, CT 06511, USA
[2]Harvard Forest, Harvard University, MA 01366, USA
[3]Department of Geosciences, University of Arizona, AZ 85721, USA
[4]Department of Geography, Center for Climatic Research, University of Wisconsin-Madison, WI 53706, USA
[5]Department of Geology and Geophysics, University of Wyoming, WY 82071, USA
[6]Earth and Environmental Science Department, Lehigh University, PA 18015, USA
[7]Department of Geography, University of Oregon, OR 89108, USA
[8]Department of Civil and Environmental Engineering and Earth Sciences, University of Notre Dame, IN 46556, USA
[9]Division of Earth and Ecosystem Sciences, Desert Research Institute in Las Vegas, NV 89119, USA
[10]Tree-Ring Laboratory, Lamont-Doherty Earth Observatory, NY 10964, USA
[11]School of Biology and Ecology, University of Maine, ME 04469 USA
[12]Department of Earth and Environment, Boston University, MA, 02215 USA
[13]Department of Geology and Geography, West Virginia University, WV 26501, USA
[14]Department of Geological Sciences, Salem State University, Salem, MA 01970, USA
[15]Southwest Climate Science Center, US Geological Survey, Tucson, AZ 85719, USA
[16]Department of Biological Sciences, University of Notre Dame, IN 46556, USA
[17]Department of Geography, University of South Carolina, SC 29208, USA
[18]Department of Geosciences and School of Natural Resources and Environment, University of Arizona, Tucson, AZ 85721, USA
[19]Biology and Paleo Environment, Lamont-Doherty Earth Observatory, NY 10964, USA
[20]Snow and Ice Data Center, CIRES, University of Colorado, CO 80309, USA
[21]Laboratory of Tree-Ring Research, University of Arizona, AZ 85721, USA
[22]Department of Biology, St. Olaf College, MN 55057, USA

*Correspondence to:* J.R. Marlon (jennifer.marlon@yale.edu)

**Abstract.** Many ecosystem processes that influence Earth system feedbacks, including vegetation growth, water and nutrient cycling, and disturbance regimes, are strongly influenced by multi-decadal to millennial-scale variations in climate that cannot be captured by instrumental climate observations. Paleoclimate information is therefore essential for understanding contemporary ecosystems and their potential trajectories under a variety of future climate conditions. With the exception of fossil

5   pollen records, there are a limited number of northeastern US (NE US) paleoclimate archives that can provide constraints on its temperature and hydroclimate history. Moreover, the records that do exist have not been considered together. Tree-ring data indicate that the 20th century was one of the wettest of the past 500 years in the eastern US (Pederson et al., 2014), and lake-level records suggest it was one of the wettest in the Holocene (Newby et al., 2014); how such results compare with





other available data remains unclear, however. Here we conduct a systematic review, assessment, and comparison of paleotemperature and paleohydrological proxies from the NE US for the last 3000 years. Regional temperature reconstructions are consistent with the long-term cooling trend (1000 BCE-1700 CE) evident in hemispheric-scale reconstructions, but hydroclimate reconstructions reveal new information, including an abrupt transition from wet to dry conditions around 550-750 CE.

5 NE US paleo data suggest that conditions during the Medieval Climate Anomaly were warmer and drier than during the Little Ice Age, and drier than today. There is some evidence for an acceleration over the past century of a longer-term wetting trend in the NE US, and coupled with the abrupt shift from a cooling trend to a warming trend from increased greenhouse gases, may have wide-ranging implications for species distributions, ecosystem dynamics, and extreme weather events. More work is needed to gather paleoclimate data in the NE US, make inter-proxy comparisons, and improve estimates of uncertainty in the

10 reconstructions.

## 1   Introduction

Many dimensions of climate – from gradual shifts in temperature and precipitation, to changing seasonal patterns, to extreme weather – influence species and ecosystem processes. An ecologically relevant understanding of regional climate variability thus requires diverse records that overlap in space and time. Such understanding would enable the investigation of long-term

15 ecosystem processes such as succession, large-scale shifts in species composition, and periodic, infrequent disturbances like droughts and flooding, using models or empirical approaches.

 In the Northeastern United States (NE US), from the Great Lakes to New England, there are few natural archives such as ice sheets, caves, or corals, that provide long regional climate records elsewhere. The last deglaciation, however, left thousands of lakes and bogs filled with sediments that have recorded local environmental changes. As a result, NE US paleoclimate has

20 primarily been inferred from local studies and vegetation studies, or from broad-scale syntheses that interpolate or downscale data to finer scales.

 Temperature changes in the NE US during the late Holocene are generally inferred from global, Northern Hemisphere, and North American research that draw on marine archives (Marcott et al., 2013), tree rings in adjacent regions (Esper et al., 2002; Mann et al., 2009; Trouet et al., 2013), and continental-scale pollen-based climate reconstructions (Viau et al., 2012; Trouet

25 et al., 2013). Such data show declining temperatures during the past 3000 years (during the "neoglacial") until about 1700 CE (250 cal yr BP), when the current global warming trend began. Sea surface temperature (SST) reconstructions from marine sediment near Nova Scotia also indicate consistently cooler-than-present conditions during the past 1500 years (Keigwin et al., 2003).

 The extent to which broad temperature trends reflect regional, terrestrial climate in the NE US is unclear. Results are contra-

30 dictory regarding changes during the Medieval Climate Anomaly (MCA; ca. 950–1250 CE; 1000-700 cal y BP) and the Little Ice Age (LIA; ca. 1400–1700 CE; 550-250 cal y BP) (Mann et al., 2009). Warming during the MCA is not observed in SST reconstructions near Nova Scotia (Keigwin et al., 2003) and ecological studies have shown complex responses to climate changes during this interval (Fuller et al., 1998; Umbanhowar, 2004). Vegetation-based climate reconstructions suggest relatively high





temperatures during the MCA in the deciduous and hardwood forests of eastern North America about 1300 years ago (Viau et al., 2012), and about 1000 years ago (Trouet et al., 2013), respectively. LIA cooling registers strongly in vegetation and fire regimes in both the Midwest and NE (Umbanhowar, 2004; Hotchkiss et al., 2007; Clifford and Booth, 2013). Vegetation is controlled by many factors in addition to climate, however, and would ideally be reserved to evaluate the ecological responses
to the climate history rather than used for both climate and ecological reconstruction, especially in a modeling context.

Data reflecting hydroclimatic variability are more abundant in the NE US, coming from tree-rings (e.g., Cook and Krusic, 2004, 2008; Pederson et al., 2014), lake levels (e.g., Shuman et al., 2002), variations in lake chemistry (e.g., Li et al., 2007), shifts in diatoms (Boeff et al., 2016), and changes in testate amoeba composition from bog sediments (e.g., Booth et al., 2012). Such data show relatively wet conditions in the late versus mid-Holocene (Digerfeldt et al., 1992; Almquist et al., 2001;
Shuman et al., 2009) with increasing effective moisture towards present (Newby et al., 2014). Episodic drought and pluvial events, documented by a variety of proxies, are superimposed on the gradual trend towards increasing moisture (Pederson et al., 2013). The synchroneity of such events is largely unexplored.

To provide a more coherent perspective on regional, terrestrial climate changes in the NE US (Peterson et al., 2013), we examine local and regional paleoclimate data from the NE US, defined as 100 to 60 east latitude, and 38 to 52 north longitude
(Fig. 1). Our objectives are to 1) identify the data resources of the region to help prioritize and motivate new data collection; 2) improve our understanding of regional climate evolution over centuries and millennia; and 3) enhance our ability to contextualize current and projected environmental changes. To these ends, we review the strengths and limitations of different kinds of paleoclimatic proxies and archives, highlighting potential connections and relationships among them. We also discuss uncertainties in paleoclimate reconstructions, an issue fundamental to the advance of climate and ecosystem modeling. Finally,
we compare and contrast the diverse NE US data sets to assess past trends in temperature and moisture availability, and to identify shared features and inconsistencies in centennial-scale events.

## 2   Review of Existing NE US Paleoclimate Records

Here we describe the most common paleoclimatic data sources (proxies) and the natural archives containing them in the NE US. Natural archives result from complex flows of materials, energy and water through the landscape (Fig. 2). As a result,
they incorporate many processes unrelated to climate. Removing non-climatic signals is thus a major challenge for studying paleoenvironmental change, and we briefly discuss the related challenges for each type of data.

Identifying records from a variety of proxies is important because different sources often reflect unique aspects of climate change. Lake level records and diatom-based records of past salinity integrate the temperature (T), precipitation (P), and evapotranspiration (ET) variations sensed by bogs with groundwater-sourced inputs (Q), operating over timescales of decades
to centuries. The sensitivity of lake hydrology to climate variability depends heavily on lake size, volume, shape, position in watershed, and degree of connectivity to subsurface water flows. Terrestrial ecological systems are governed by climatic variability and ecological responses to this variability at timescales of years to centuries, and at levels of ecological organization ranging from individual organisms to communities and ecosystems. Proxies obtained from tree rings sense the growth responses



of individual trees at timescales of seasons to decades, while the biomarker, isotopic, and pollen signals recorded in bog and lake archives are typically time-averaged by depositional processes, with the degree of time-averaging varying among archives. The proxies found in marine archives are mostly produced by marine organisms (e.g. biomarkers) but some such as pollen are terrestrially sourced (Fig. 2).

Each natural archive is associated with a typical record length (Fig. 3), which in turn constrains the timescales of climate variability that can be reconstructed (e.g., hourly, seasonal, decadal, millennial). Longer records tend to have lower temporal resolution due to sampling constraints. The diversity of climatic and non-climatic signals that register in a proxy, along with the differences in record lengths and spatial coverage makes it difficult to compare records and presents a key challenge for building robust paleoclimate reconstructions. Few records can capture multi-decadal temperature or moisture variability with

high temporal resolution (e.g., seasonal or annual variations). Nonetheless, an assessment of the existing data in the NE US highlights opportunities for bridging multiple scales of variability, such as paired records from nearby bogs and lakes, or composite records from varved lakes and regional tree-ring reconstructions.

## 2.1   Modern and Historical Climate Records

Observations and measurements of weather and climate during the modern (1950-present) and historical (1600–1950 CE)

periods in the NE US provide accurate, comprehensive information about recent climate changes. Hourly temperature and precipitation measurements, relative humidity, wind speed and direction, and barometric pressure are measured at various elevations. Other variables such as soil moisture can be calculated. Hourly or daily data can be aggregated seasonally, annually, and decadally to examine short-term variability. In addition to documenting ongoing climate changes, such data support modeling future changes (Hayhoe et al., 2007) and the calibration of paleoclimate records.

The United States Historical Climatological Network (USHCN), which includes hundreds of US weather stations that are often still in operation today (http://www.ncdc.noaa.gov/oa/climate/research/ushcn/), provides continuous climate data spanning the past century or more through to present. Historical Fort Data from the National Archives and Records Administration (http://mrcc.sws.uiuc.edu/FORTS/inventory1.jsp) extends back to the 19th century in some cases. Some of these stations became part of the USHCN, thereby providing a continuous record of climate from the historical period through the present

(Tipton et al., 2016). Hiatuses are common in the Fort data however, and measurements were not taken at the same time of day at different stations, so uncertainty is generally higher than in the USCHN data.

A variety of other historical climate records are used to reconstruct particular weather events, such as storms and hurricanes (Mock, 2012). Documentary archives generally span decades or a few hundred years, and come from early instrumental records, personal diaries, newspapers, and ship logs. Such records can be very accurately dated and detailed, but spatial coverage is

limited and uncertainties tend to increase with record length. Other sources of historical data come from careful, systematic observations of environmental changes like ice-out dates (Hodgkins, 2013), changes in the timing of seasons or precipitation patterns, and changes in lake levels or phenology (e.g., arrival date of particular migrating bird species, or the blossom dates of certain plant species) (Primack et al., 2004).



Modern and historical data show annual temperatures in the NE US increasing by almost 1.1°C and precipitation increasing by about 12 cm (more than 10%) over the past century (Horton et al., 2014). Precipitation timing is also changing, with the heaviest events from 1958-2012 increasing 37% in the Midwest, and 71% in the northeast (Karl, 2009). The frequency of daily summer precipitation (June, July, August) has significantly increased since the mid 1990s in the Ohio Valley of Ohio

and Kentucky and in eastern New York State (Bishop and Pederson, 2015). Seasonal changes are also evident, with warming strongest in winter and precipitation increases most pronounced in the Fall (Kunkel et al., 2013). Loss of winter ice cover on Lake Superior has caused the summer surface water temperatures there to increase approximately 2.5°C between 1979–2006 (Austin and Colman, 2007). In Maine, ice-out dates (the dates when lake ice first breaks-up each spring), are getting earlier, reflecting warming over the past century (by 0.6 days/decade from 1884 through 2008) (Hodgkins, 2013). Sebago Lake in

Maine has been failing to freeze completely at an increasing rate since 1937, and projections for continued warming signal significant ecological and economic impacts in the region through its effects on fisheries, transportation (e.g., snowmobile trails) recreation, and other resources. Warming thus appears to be more rapid in southern than northern New England thus far, but upwards trends in both temperatures and precipitation are expected to continue in coming decades (Anderson et al., 2010).

## 2.2   Tree-ring Records

Tree-ring records provide the most abundant data about seasonal to decadal drought and temperature variations beyond the historical record (Speer, 2010). Records from long-lived species like bristlecone pine (*Pinus longaeva*) or baldcypress (*Taxodium distichum*) are used to extend instrumental records of climate back 1000-3000+ years in some cases, but limited tree longevity and preservation of samples for cross-dating often limit longer-term analyses. . Using ring widths, wood density, or stable isotopic content, dendroclimatic (tree-ring based paleoclimate) reconstructions of temperature and moisture availability can

be created from a network of tree-ring records. In addition, growth suppression (Trouet et al., 2016) and tree-ring anatomical variability (Therrell and Bialecki, 2015) can be used to reconstruct extreme events, such as hurricanes and floods.

A strength of dendroclimatology hinges on precise and accurate dating of each ring (Black et al., 2016) – a non-trivial exercise, particularly in closed canopy forests where many factors influence growth. The approach can identify and date specific extreme climatic events or decadal climate fluctuations. A second strength of tree-ring reconstructions is the direct calibra-

tion and verification of the climate-growth model from instrumental data. Rigorous statistical testing of these models builds confidence that reconstructions of historical climate represent an approximation of the range of past variation, or at least of variations that do not kill the trees. The methods in which climate are reconstructed from tree rings are reviewed in detail in Cook and Kairiukstis 1990. Key limitations of tree-ring records of past climate are predicated on tree longevity, the ability to retain low-frequency trends, and the geometric and ecological factors influencing stem growth (Fritts, 1976; Cook and Peters,

30   1997).

In the NE US, tree-ring records are limited in length, rarely extending beyond the past 500 years. Growth is influenced by non-climatic and climatic factors (Cook and Pederson, 2011), and most NE US trees are moisture- rather than temperature-sensitive (Stahle and Hehr, 1984; Cook, 1991; Meko et al., 1993). Conifers at high elevations or northern range margins have the strongest significant and positive relationships to temperature (Conkey, 1982; Pederson et al., 2004; Bhuta et al., 2009),





and the temperature sensitivity of red spruce wood density (Conkey, 1986) and Atlantic white-cedar ring widths (Hopton and Pederson, 2005) in particular have produced reliable reconstructions. Red spruce sensitivity to acid rain, however, may complicate the most recent temperature reconstructions (Webster et al., 2004).

In the NE US, only a few tree species in a specific portion of their range are sensitive enough for temperature reconstruc-
tions (Conkey, 1986; Pederson et al., 2004; Hopton and Pederson, 2005). Several records 1000 years or longer are solely from cliff-dwelling northern white cedar or eastern red cedar, and account for a lower proportion of annual variance in the instrumental record versus most other reconstructions. Such limitations can be overcome by combining multiple species (Cook and Pederson, 2011; Maxwell et al., 2011; Pederson et al., 2013), but this limits the robustness of longer reconstructions.

The most common paleoclimatic reconstruction target from tree rings is the Palmer Drought Severity Index (PDSI), espe-
cially in the NE US due to the limited temperature sensitivity of most species. The North American Drought Atlas (NADA) (Cook and Krusic, 2004; Cook et al., 2004) provided new insights into broad-scale hydroclimate variations. Most early NADA research focused on the great droughts of the western US and Great Plains (e.g. (Fye et al., 2003; Cook et al., 2004; Stahle et al., 2007), but within the last decade, a focus on NE US hydroclimate has emerged. In the central Mississippi River Valley, a summer PDSI reconstruction shows a sustained wetting trend from the 1100-1247 CE dry epoch through present day (Cook
et al., 2010). Analysis of tree rings from the Red River indicate an extended dry period from 1675 to 1770 CE (George and Nielsen, 2002). Summer moisture across the central portion of the eastern US has been increasing since at least the early 1800s (McEwan et al., 2011). Similar trends are evident from PDSI reconstructions in eastern N.Y. state and western New England, with a general reduction in the severity and duration of drought in the 20th century versus the prior 350 years (Pederson et al., 2013). At the northern end of the Mississippi River in Minnesota, Iowa, and Wisconsin, summer PDSI over the last 500 years
also indicates a general wetting trend (Figure 3) (Pederson et al., 2015).

At annual to decadal time scales, properly dated tree-ring records of past climate demonstrate how climate affects ecological and societal dynamics (Stahle et al., 1998; Cook et al., 2010). Both qualitative and quantitative tree-ring data have been used to understand how intensive drought contributed to the severe mortality rates among the first European settlers in Jamestown, for example (Stahle et al., 1998), and how paleoclimate variations can affect ecosystem services today (Pederson et al., 2015).

## 2.3 Bog Records

Analyses of peatland sediment from bogs have been widely used in North America and Europe to provide information about changes in vegetation communities and fire regimes from fossil pollen and charcoal. Bog sediment from ombrotrophic, or raised bogs, are also a unique source of hydroclimate data, however, because precipitation is their sole moisture input. Bog surface-moisture responds predominantly to growing season precipitation (Booth et al., 2010; Charman et al., 2004; Charman and Blundell, 2007; Charman et al., 2009), which affects the community composition of testate amoebae in the bog. Testate
amoebae are a group of protists that produce decay-resistant shells. Different species have different moisture sensitivities, and so measuring changes in relative taxa abundances (i.e., community assemblages) over time enables the reconstruction of past changes in water-table depth (Booth et al., 2002; Booth, 2008; Mitchell et al., 2008).





Comparisons between water-table depth reconstructions and PDSI show strong correlations (Booth, 2010), as do downcore comparisons with hydrologic and climatic data (Booth et al., 2010). Calibration datasets for testate amoeba are based on samples collected along moisture gradients in the bog, however, which means that reconstructions of absolute precipitation values is not possible. As a result, inferred water-table depth changes at a coring location reflect relative rather than absolute

changes in surface moisture through time.

For water-table depth reconstructions, a weighted-average model is developed from a calibration dataset of modern testate amoebae communities is used to infer water-table depths from compositional changes (Booth, 2008). Typically, additional analyses are done to evaluate the robustness of the paleohydrological reconstruction. For example, it is common to compare 1) the testate amoebae reconstructions to peat humification data from the same core and 2) records among bogs from the same

region. These comparisons provide an assessment of both the robustness of the paleohydrological reconstruction as well as its climatic sensitivity.

Modelling studies suggest that extreme events register strongly in bog records (Morris et al., 2015; Clifford and Booth, 2015), whereas lower magnitude and lower frequency changes in water-table depth may be more strongly influenced by autogenic processes such as lateral peat-bog expansion, vertical peat growth, decomposition-induced changes in hydraulic conductivity,

and other ecological dynamics (Charman et al., 2006; Swindles et al., 2012; Morris et al., 2015). As in dendroclimatic reconstructions, detrending WTD data removes potential non-climatic signals, but concurrently limits paleoclimate inferences to extreme centennial-to-multidecadal events (Charman et al., 2006; Booth et al., 2012).

In eastern North America, the best-dated sites thus far are from Michigan and Maine (Booth et al., 2012; Clifford and Booth, 2013). Drought, vegetation, and fire reconstructions from several New England records reveal an abrupt decline of eastern

hemlock (*Tsuga canadensis*) and American beech (*Fagus grandifolia*), concomitant with an increase in pine (*Pinus* spp.) and oak (*Quercus* spp.) between 1350-1450 CE (500 and 600 yr BP) from increased drought and fire (Clifford and Booth, 2013).

## 2.4   Lake and Marine Records

Lake and marine sediments can capture long (multi-decadal to millennial-scale) variations in climate, and several methods used to study paleoclimate in sediments can be applied to both lake and marine sediments, including fossil pollen and approaches

based on organic geochemistry. Lake and marine sediments are commonly analyzed for a broad range of environmental data, some of which have paleoclimate information, but many of these data sources are too complex to provide consistent, reliable data from a variety of sites. Grain size analyses, magnetic susceptibility, and loss-on-ignition, for example, are commonly quantified in sediments but often provide information that is ambiguous in terms of its climate signals.

Important differences exist in reconstructions from marine versus lacustrine sediments. The different organisms and deposi-

tional contexts affect the kinds of variables that can be reconstructed, the typical record lengths (marine records are longer), the typical temporal and spatial resolution (more coarse for marine sediments), and the amount of uncertainty in the dating. The number and nature of adjustments in radiocarbon dating of sediments varies, for example, in lake versus marine sediments. In addition, influences from non-climatic factors also differ between lake and marine records. For example, sediment deposition



is affected by oceanic currents in marine contexts, and from inflowing and outflowing streams or from ground water flow in a lake basin.

### 2.4.1 Annually Laminated Sediments (Varves)

In some lakes, the annual sediment accumulations form distinct annual layers, called varves, that are often visible as thin,
differently-colored bands. This typically occurs in small lake basins with steep slopes and limited wind-driven currents that work to prevent seasonal mixing of surface and bottom waters. In this case, the deep benthos is unventilated and hence permanently anoxic, preventing organisms that normally exist in the sediment from disturbing the layering. Varved lakes are rare, but when found they can yield unusually detailed information about annual and sometimes seasonal variations in environmental conditions over millennia.

Paleoclimate information from varved lakes has generally come from the interpretation of changes in vegetation composition from fossil pollen (described below) (Gajewski, 1987). A sediment record from Elk Lake, Minnesota with distinct annual layers (varves), shows that white pine (*Pinus strobus*) largely replaced oak savanna about 750 BCE (2700 cal yr BP) (Bradbury, 1996), for example. Variations in the thickness of the varves themselves, however, can also provide climate information. Varve thicknesses from Elk Lake, Minnesota, were linked to solar activity and wind intensity during the early and mid-Holocene,
but late-Holocene varve thicknesses were not easily explained (Bradbury, 1996). Pollen and charcoal data from Hell's Kitchen Lake in north-central Wisconsin, for example, indicated alternating moist and dry intervals over the past 2000 years, but yielded no long-term trends in temperature or moisture (Swain, 1978). Data from (Hubeny et al., 2011) are used here (Fig. 1) to show variations in varve thickness from two sites in New York and Rhode Island; varve thickness variations in both records were positively correlated with hydroclimate indices that suggest broad-scale forcings.

### 2.4.2 Lake Level Reconstructions

Changing lake levels provide a clear and visual indication of changing climate conditions – specifically of variations in moisture balance integrated over decades or longer. In the western US, for example, former strand lines of pluvial lakes Bonneville and Lahontan testify to wetter conditions in the distant past that are difficult to even imagine based on the modern climate (Fritz, 1996). In humid regions such as the NE US, such evidence may not be as visually obvious because high water now submerges
ancient shorelines from past dry periods (Freeman-Lynde et al., 1980; Hutchinson et al., 1981; Mullins and Halfman, 2001). Shoreline features have long been used to reconstruct lake-level changes in response to drought events, but records based on discrete features are not continuous.

More recently, lithologic and magnetic measurements made from sediments have been used to develop continuous reconstructions of past drought and periods of sustained aridity. For example Isothermal Remanent Magnetization (IRM) was used
to produce a record of Holocene drought events (Li et al., 2007) from White Lake, New Jersey, that are strongly correlated with cold events in the North Atlantic (Bond et al., 2001). The IRM records show large shifts in lake levels that represent discrete drought events; gradual trends are not evident in the data (but see Geiss et al. (2003). Strong magnetism in the sediments occurs when the sediment is comprised of marls (lime-rich sediment) rather than gyttja (organic-rich sediment). When the lake levels



are reduced, the marls are exposed, oxidized, eroded, and transported to the deeper sediments of the lake, causing both a color change from brown to light yellow and a corresponding increase in magnetism, although the origin and processes forming the color change and the magnetism are different (Li et al., 2007). Sediment analyses that produce paleoclimatic records can sometimes come from sites with unique hydrological or bathymetric characteristics that make them sensitive recorders of climate

change, but that are not reproducible at other sites; such records are helpful for reconstructing local hydrological variations but local calibration data and/or other nearby proxies are thus needed to understand the regional climatic signals embedded in these lake-level reconstructions and evaluate their uncertainty.

Several lake level reconstructions exist in the NE US based on ground penetrating radar (GPR) and transects of multiple sediment cores from a lake margin; this approach enables continuous reconstructions of shoreline advances and retreats

through time that at least partly reflect changing hydroclimate (Pribyl and Shuman, 2014; Digerfeldt, 1986). Factors such as isostasy and tectonics can influence large lakes (Balco et al., 1997, 1998), but operate on scales greater than the small ponds that have been used to reconstruct regional hydrologic changes (Newby et al., 2014). Groundwater flow directly controls the water levels in many small ponds and lakes(Almendinger, 1993; Winter, 1999), but annual changes in precipitation and evaporation have strong regional impacts (Weider and Boutt, 2010). Suitable lakes for lake-level reconstructions are usu-

ally formed within surficially-closed basins and are controlled by climatically-sensitive, near-surface groundwater fluctuations within well-defined watersheds. In the NE US, such lakes are ubiquitous, and comparisons of multiple lakes can limit confounding variables (Newby et al., 2014). Moreover, because lake-level reconstructions are based on the changes in physical properties of the sediments themselves (i.e., direct measurements of past shoreline locations), biological and geochemical processes do not complicate interpretations. Care must be taken to consider hydrologic complexities (Donovan et al., 2002; Smith

et al., 2002; Steinman and Abbott, 2013), but multi-site validation using lakes located high in their local watersheds and in different hydrologic settings can improve climatic interpretations (Newby et al., 2014).

Dating is the primary constraint on the temporal resolution of lake level reconstructions because sediments must be dated using radiocarbon methods, which introduces a range of uncertainties from sampling, dating, and calibration (Newby et al., 2014). As a result, lake level reconstructions typically identify centennial to millennial-scale climatic variability, although

sediment signals of decadal droughts, such as the 1960s drought in New England have been detected and dated (Newby et al., 2009). In addition, because lake-levels vary in response to continuous changes in precipitation and evaporation usually integrated over 1-5 years (Mason et al., 1994), sub-annual hydroclimatic variations cannot be directly inferred (Shuman and Donnelly, 2006).

Transects of sediment cores are used to identify shifts in sand-mud boundary locations at the margin of each lake's littoral

zone (Digerfeldt et al., 1992). GPR or other geophysical datasets can confirm the geometry of such changes (Newby et al., 2009; Shuman et al., 2009). A transect of two or more sediment cores across the lake are taken based on the GPR results. The results are then used to identify boundary shifts from changes in sediment characteristics, which enable the reconstruction of lake level variations over time.

Each centimeter (or finer) of sediment in a lake cores is classified as organic-rich profundal or as sand-rich littoral based on

loss-on-ignition, bulk density, or x-ray fluorescence (XRF); the information is then used iteratively in an algorithm that creates



a continuous estimate of lake level through time (Pribyl and Shuman, 2014). Reconstructed lake level variations can be coupled with watershed size and related information to reconstruct estimates of effective moisture (Shuman et al., 2009; Newby et al., 2014), or the ratio of actual evapotranspiration to potential evapotranspiration. In central New England, records are now being constructed for the creation of time-series of effective moisture at centennial time scales throughout the Holocene (Marsicek 5 et al., 2013; Pribyl and Shuman, 2014).

Because lake level variations represent century or longer-term fluctuations, these records are difficult to calibrate against modern observations (although see Newby et al., 2009). Modern calibration work has therefore focused on understanding water depth-sediment relationships rather than on the calibration of the reconstructed and observed precipitation minus evaporation (P-E). Efforts to validate the reconstructions include comparisons between lake-level changes and pollen inferred moisture 10 changes (Marsicek et al., 2013) and comparisons of the reconstructions from multiple, closely located lakes (Newby et al., 2014). (Plank and Shuman, 2009) also digitized hundreds of aerial photos of lakes from the 1930s to examine how their levels changed relative to PDSI observations, and more information about the hydrologic responsiveness of lakes to drought could be gleaned from such data. Newby and colleagues Newby et al. (2009) identified a thin sand layer in the upper portion of a core from New Long Pond, in Plymouth, Massachusetts, that dated to the 1960s drought, which depressed the local water table by 15 >3 m; the sand layer shared the characteristics of many large sand layers used to infer prehistoric changes in water level. It may be possible to quantify the magnitude of the drought from the sediment data and compare it with the water-table observations in future research. Recent work in New England suggests that sustained drought may have occurred in the past despite limited changes in moisture availability in the more recent past (Newby et al., 2014).

### 2.4.3 Organic Geochemistry

20 The geochemical composition of bulk organic matter in the sediments has long been used for environmental reconstruction. There are a wide array of geochemical data that can be gathered from this organic matter, including abundance information, elemental distributions, and stable isotope ratios. With an understanding of thermodynamics, fractionation of stable isotopes, and advances in instrumentation applications, uses of stable isotopes have continued to grow. In marine sediment archives, stable oxygen ($\delta^{18}O$) and carbon ($\delta^{13}C$) isotope compositions of carbonate fossilized foraminiferal tests are used to reconstruct 25 SST and global ice volume and dissolved inorganic carbon composition of ocean water, respectively (Ravelo and Hillaire-Marcel, 2007). Likewise, the $\delta^{18}O$ and $\delta^{13}C$ values of lacustrine material has been examined as a source of paleoclimate information (Leng and Marshall, 2004; Meyers, 2003). These geochemical analyses are routine in settings with excellent preservation, however the archives can be degraded or represent mixed terrestrial and aquatic inputs (Castañeda et al., 2011), complicating or overprinting the original signature.

30 Biomarkers are organic molecules, compounds or groups of compounds that can be linked to specific source organisms or groups of organisms, and are often recovered in locations where other bulk proxies are not available (Castañeda et al., 2011); they are increasingly used for paleoclimate information from lacustrine and marine settings. These chemical signatures are insoluble in water, often chemically inert with low volatility, and resistant to biodegradation, making them long-lived in the environment and well suited for studies of the past (Eglinton and Hamilton, 1967; Eglinton and Eglinton, 2008).





Biomarker proxies of temperature from aquatic settings, $U^K_{37}$, based on differing number of double bonds of alkenones from haptophyte algae (Brassell et al., 1986; Herbert, 2001; Prahl and Wakeham, 1987), and $TEX_{86}$, based on the membrane lipid ring structures of aquatic Thaumarchaeota (Powers et al., 2004; Schouten et al., 2002), have been more recently used for temperature reconstructions from lakes and oceans. Modern calibration studies and comparisons between these proxies,

along with foraminiferal reconstructions when available, have revealed important differences in the seasonality (e.g. (Pitcher et al., 2011)) and depth of production of the organisms these proxies are based on e.g (Karner et al., 2001), with significant implications for use of these proxies for SST reconstruction.

Terrestrial plant leaf wax biomarkers have been used to study hydroclimate using compound specific $\delta^2H$. Derived from the surface coating on leaves, leaf wax compound specific hydrogen isotope values ($\delta^2H$) demonstrate strong relationships

with mean annual precipitation (Sachse et al., 2004), though new research suggests they reflect a more limited interval of time surrounding leaf development (Tipple et al., 2013). Leaf wax compound $\delta^2H$ values have been used to make a range of paleohydrological reconstructions (Sachse et al., 2012). In the NE US, Nichols and Huang (2012) used the ratio of the $C_{23}$ n-alkane (representing bog-dwelling Sphagnum) to the $C_{29}$ n-alkane (representing 5 dry-tolerant vascular plants) as an indicator of relatively wet versus dry conditions over the Holocene at a coastal bog in Maine. Hou et al. (2007, 2006) developed

paleoclimate reconstructions from $\delta^2H$ values of the aquatic lipid behenic acid ($C_{22}$ n-alkanoic acid) from a sediment core in Blood Pond, Massachusetts, and showed that variations in local climate paralleled those observed in the $\delta^{18}O$ variations from the Greenland Ice Sheet Project 2 (GISP2). Though the resolution is extremely low for the last 3000 years, $\delta^2H$ of long chain terrestrial leaf wax compounds from Crooked and Berry Ponds, Massachusetts, suggest a wetting trend between 3-2 cal ka before becoming more dry and stable after 2 cal ka (Shuman and Donnelly, 2006).

Several other Holocene climate reconstructions from the NE US have been developed from compound-specific $\delta^2H$ in lake sediments (Hou et al., 2006; Huang et al., 2002), but these have primarily been used as corroborating evidence for large, abrupt climate changes during deglaciation, such as the 8.2 ka event (Hou et al., 2007).

As with other proxies, there are uncertainties, including the precise factors that influence lipid $\delta^2H$ values (Sachse et al., 2012) and specific organisms responsible for generating the lipids used in $TEX_{86}$ (Schouten et al., 2013), these approaches

offer novel and promising ways to reconstruct past hydrologic and temperature variability, respectively.

### 2.4.4 Fossil Pollen

The most abundant paleo data in the NE US are pollen records developed from lake sediments that have been used primarily to understand changes in forest composition (Williams et al., 2004; Blois et al., 2011; Oswald and Foster, 2012; Marsicek et al., 2013). Through the use of transfer functions, pollen data have also been used to reconstruct paleoclimate (Webb et al., 1987,

1993; Viau et al., 2006; Bartlein et al., 2011; Viau et al., 2012).

Climate can be reconstructed from fossil pollen in sediment records because a) the geographic distribution and abundance of plant species is strongly influenced by climatic factors such as temperature minima, temperature maxima, and water availability (Woodward, 1987) and b) pollen abundances and distributions are a reliable, albeit biased, indicator of plant abundances and distributions (Webb, 1974; Solomon and T. Webb, 1985). Although pollen grains are rarely identified to the species level, higher



taxa distributions are also determined by climate, and relationships can therefore be constructed between taxa assemblages and their locations in climate space (Bartlein et al., 2011). In the NE US, many species identifications have been made from pollen data, which can refine the environmental interpretation (Lindbladh et al., 2003). Spatially-extensive surface pollen sample data sets (i.e. modern pollen datasets) are typically used to determine the contemporary relationships between climate

and plant assemblages (Williams et al., 2006; Bartlein et al., 2011). Such training data are then used to establish empirical relationships between taxon abundance and selected climate variables. The unique relationships between specific taxa and climate, as determined by the analysis of the modern spatial datasets, are then applied to make paleoclimatic inferences from temporal variations in fossil samples. Such "space for time" approaches use a variety of algorithms for transferring spatial climate-pollen relationships to temporal pollen trends, such as modern analogue, constrained analogue, response surfaces, and

transfer functions (Telford and Birks, 2009). Modern species distributions or process-based (inverse) modelling can also be used to predict vegetation composition, particularly in terms of plant functional types (Wu et al., 2007; Garreta et al., 2010). In the latter case, both species distributions and surface samples can be reserved for validation testing (Bartlein et al., 2011).

The use of fossil pollen records to reconstruct climate has a long history in eastern North America (Bartlein and Webb, 1985; Gajewski, 1988; Prentice et al., 1991; Webb et al., 1993, 1998; Jackson et al., 2000; Williams et al., 2000; Bartlein et al., 2011;

Viau et al., 2012; Wahl et al., 2012). Pollen data have been used to reconstruct climate variables such as mean temperature of the coldest month, mean temperature of the warmest month, mean annual temperature, and mean annual precipitation and bioclimatic variables. Such reconstructions quantify the changes that have often been observed in the pollen data itself. Records from the prairie-forest ecotone in Minnesota, Iowa, Wisconsin, and farther west, for example, show rapid prairie expansion in the early Holocene and slower subsequent afforestation in response to wetter conditions (Williams et al., 2009). Widespread

expansions of taxa adapted to cool, moist climates, like spruce and fir, are documented by pollen data from the late Holocene (Gajewski, 1987). Oak, which is drought-tolerant and adapted to periodic disturbance, was reduced, and moisture-dependent chestnut (*Castanea dentata*) increased after ca. 1000 BCE (2950 cal yr BP) in New England (Davis, 1980; Shuman et al., 2004).

Short-term variations in plant assemblages associated with disturbance and succession often cannot be resolved in pollen

records, however, due to resolution constraints, but centennial to millennial variations generally track climate changes with little lag time (i.e., less than 100 years) (Williams et al., 2002). An important constraint in using pollen data to reconstruct temperature variations exists when vegetation distributions may have been highly disturbed by human activity (e.g., for agriculture).

Another important caveat holds for the use of pollen data in climate reconstructions that are supporting modeling efforts.

If pollen data are used to reconstruct vegetation changes in ecosystem models, then the same data cannot also be used to reconstruct the climate changes. Using the same data to drive both vegetation and climate change could result in circular reasoning, because climate changes are expected to drive the vegetation changes in the first place.



## 2.5 Diatoms

Sediment diatoms provide another hydroclimate proxy, because diatom species have different salt tolerances, so community composition changes with shifts in lake salinity, or temperature and/or stratification (Fritz et al., 2000; Rühland et al., 2015; Saros et al., 2012). As with most proxies, diatoms community composition also responds to non-climate factors such as lake

location and morphometry, which affect lake circulation, for example (Boeff et al., 2016; Wigdahl-Perry et al., 2016). Many of the space-for-time transfer functions used in pollen-based paleoclimatology have also been applied to diatoms (Juggins and Birks, 2012). Although diatoms have not been analyzed as often as pollen in the NE US, they offer a unique opportunity to disentangle environmental changes due to human activities from those of climate change (Battarbee et al., 2012). More comprehensive discussion of diatom responses to climate may be found in Saros et al. 2012, 2015 and Ruhland et al. 2015.

## 10  3  Coherence of NE US Climate Change Records

### 3.1  Methods

To assess the coherence of climate changes among the diverse paleoclimate records in the NE US, we smoothed the high-resolution records, and composited and smoothed the hundreds of pollen records in the NE US from the Neotoma Paleoecology Database (http://neotomadb.org). We identified drought events in the testate amoeba records and calculated sums of drought

events from these events to make this proxy comparable to gradual changes in lake levels and other hydroclimatic data.

High-resolution records, including the tree-ring based temperature reconstruction from Esper et al. 2002, the composite varve thickness record from Hubeny et al. 2011 (created by merging data from two sites along the Atlantic margin), and the pollen-based temperature and precipitation records, were all smoothed using the same approach. The data were first binned (not interpolated) into regular time steps (5-yr bins for all but the pollen data, which were binned into 10-yr intervals due to their

lower average temporal resolution). Once binned, the data were smoothed with a 50-year half-window width using locally-weighted regression, robust to outliers (Loader, 2007)), except for the pollen data, which were smoothed using a 100-year half-window width. The window widths were chosen to optimize the identification of both trends and shared major features in the records on centennial to millennial time scales. Bootstrap confidence intervals are created by sampling the sites in the full group (with replacement) and re-running for 1000 iterations.

For the pollen data, the modern analogue technique was used to reconstruct climate variables (Williams et al., 2011). In this analysis, the climate variables were converted to anomalies so that shared patterns could be observed regardless of differences in the absolute values across the NE broadly. Pollen records were compiled from sites between 40° and 50° latitude and -70° and -100° longitude (Fig. 1b).

Seven testate amoeba records (Booth et al., 2012; Clifford and Booth, 2013) were decomposed into low and high frequency

components so that reconstructed drought events could be separated from trends in water table depth that may not be related to climate changes (e.g., a gradual growth of the bog may cause shifts in water table depth that do not reflect shifts in atmospheric moisture). CharAnalysis (Higuera et al., 2009) was used to interpolate and decompose the water table depth variations into





low frequency trends and discrete events that represent periods of drought. A constant of 10 was added to the South Rhody record in order to ensure that all water table depth changes were positive values. Smoothing windows of 200 years were used to separate the low frequency trends from the events for all sites except Irwin Smith Bog, which required a 300-year smoothing window. Residual values (CHAR-Background CHAR) were smoothed using a lowess smoother robust to outliers

in order to identify drought events. The drought magnitudes (increase in water table depth $cm^2 \; yr^{-1}$) were plotted and drought frequencies were summarized using a kernel density estimator (Venables and Ripley, 2002) (200-year bandwidth) to identify long-term variations between wetter and drier periods.

Tree ring data from the North American Drought Atlas (NADA) (Cook et al., 2010) were used to calculate the Drought Area Index (DAI) for the north/central and eastern portions of the NE, and for both combined. Grid cells included in the north/central

region are from -100° to -90° longitude and 37.5° to 52.5° latitude, and from 37.5° to 47.5° latitude and -80° to -65° longitude for the eastern region. Data in each sub-region (Midwestern and North Atlantic Coast) were smoothed using methods from Cook et al. (2004).

To characterize the broad-scale spatial and temporal trends in temperature from the last century across the NE US, we analyze data from the Global Historical Climatological Network (GHCN) using empirical orthogonal functions (EOF). We

also use EOF analysis to identify the spatial structure in the NADA from 1700-2005.

Two reconstructions of New England storms, including tropical cyclones and hurricanes, were created from the regional to synoptic scale historical data. For tropical cyclone candidates, characteristics were analyzed to ensure that a storm was indeed tropical (Mock, 2008). Records clearly exhibiting cold frontal passages, lack of an inner warm-core system, and falling V-shaped bar graphs were classified as extratropical, but all pre-1871 records were noted if they had winds gusting to gale

or hurricane force during hurricane season. Data for each storm were also examined through descriptions of storm surge and damage to assess storm intensity consistent with the Saffir-Simpson scale.

## 3.2 Uncertainty in Paleoclimate Reconstructions

There are multiple sources of uncertainty in paleoclimate reconstructions stemming both from natural spatial and temporal variability as well as from our ability to understand, measure, and model environmental processes (Evans et al., 2013). Iden-

tifying, quantifying, and assessing the importance of different sources of uncertainty is a growing challenge in paleoclimate science as new proxies and models proliferate and as new techniques for data-model assimilation place increasing importance on accurately quantifying uncertainty (Crucifix, 2012). The examination of uncertainty can occur through comparisons among different collections of observational evidence, as well as between models and data. Providing quantitative estimates of uncertainty requires some kind of calibration, or a comparison between the paleo reconstruction and independent data source with

known accuracy.

Uncertainty stems from two key sources of variability: 1) natural variability, which occurs through space and time in ecological and climatic processes that may be stochastic or systematic, but are ultimately controlled by external factors; and 2) our knowledge of such changes, which can result from limitations or errors in observation, sampling, measurement, dating, analysis, and other sources (Goring et al., 2012; Tingley et al., 2012). Uncertainty can be examined qualitatively or quantitatively,





and both approaches are common in paleo research. For established climate proxies like tree rings, quantitative calibration is common practice, and typically includes a combination of calibration uncertainty, sample depth-related uncertainty, and dating uncertainty (Esper et al., 2007; Trouet et al., 2009). Many other types of data (e.g., from lake sediments) contain paleoclimate information that cannot be directly calibrated, either because modern data for the variable measured do not exist

or do not align closely enough with the temporal and/or spatial scale of climate variability that is being reconstructed. In lake level reconstructions for example, modern and historical precipitation records generally do not extend back far enough in time to allow validation of the centennial and millennial-scale variations in the paleodata. In this case, inter-site and inter-proxy comparisons can still be used to increase confidence in the data. A comparison between two lake-level reconstructions from nearby sites in Massachusetts (Fig. 5A) for example, shows a positive correlation of 0.654 ($R^2$), suggesting that the trends

reflect shared climate variations. Another approach is to couple lake-level data with information about watershed hydrology to calculate/model variations in P-E at a site, and compare the results to P-E reconstructed from pollen data extracted from the same site (Fig. 5B); at Deep Pond, Massachusetts, the comparison yields an $R^2$ of 0.427. Such analyses can reduce uncertainty by allowing assessments of relative changes over time; they can also serve as qualitative benchmarks for models. However, reducing uncertainty, whether through cross-site or cross-proxy comparisons, through careful site selection, radiocarbon dating

and other methods, is insufficient to provide the quantitative estimates of uncertainty required for rigorously assimilating data into models.

A key step in paleoclimate reconstructions based on changes in biological community compositions is determining the relationships between modern spatial distributions of species (or taxa) and modern climate conditions. The space-for-time calibration (described above) is illustrated in Fig. 5C, where modern (instrumental) measurements of annual precipitation

are compared against reconstructed annual precipitation from modern plant assemblages inferred from pollen data in lake sediments (Marsicek et al., 2013). Challenges arise however when no modern analogue exists for a particular combination of plant taxa, making the past climate difficult if not impossible to reconstruct for certain time periods and locations (Jackson and Williams, 2004).

Calibrating paleoclimate information from changes in the community composition of testate amoeba follows a similar space-

for-time approach. Variations in past taxa assemblages are used to reconstruct changes in water table depth, which is an indicator of precipitation. Changes in community composition, however, must first be linked to variations in water table depth, which is also based on the space-for-time substitution approach, and in this case allows the determination of the relationship between modern community composition across modern climate and environmental spatial gradients. Booth (2008), for example, developed and tested a variety of transfer functions between testate amoeba communities and water table depths by comparing

the spatial patterns in taxon distributions from eastern North America and the Rocky Mountains against modern observations of water table depth at each site using several approaches (Fig. 5D). The transfer functions can then be applied to temporal variations in testate community composition down the core. The accuracy of alternative transfer functions can be evaluated using cross validation, spatial leave-one-out approaches, and similar statistical methods. Bayesian approaches to calibration can help improve prediction skill (Tingley and Huybers, 2009), and may be applicable to other proxies that require transfer

functions, such as reconstructing climate from plant assemblages using pollen data.



Ideally, paleoclimate reconstructions can be quantitatively calibrated using modern climate data. Modern PDSI measurements for example, are often compared against reconstructions of PDSI from variations in tree-ring widths. A moderately strong correlation ($R^2 = 0.547$) is observed between instrumental versus reconstructed PDSI values from tree-ring widths in New York over the period of overlap 1895-2000 CE (Fig. 5E). Such analyses allow the calculation of error estimates for drought

reconstructions for the preceding centuries (Pederson et al., 2013). In a similar example from Booth (2010), water-table depth (WTD) was reconstructed for Hole Bog (Booth, 2008), and these measurements were used to infer variations in past drought events. The correlation between WTD estimates and (averaged) instrumental PDSI values for 1895-2003 CE from the broader climate division ($R^2 = 0.39$) helps assess but not partition the uncertainty.

To understand clearly whether differences in observed versus predicted data result from natural spatial or temporal variability

in climate, or from some aspect of the measurement process (e.g., differences in the locations of the proxies, or in sampling error), process models can be developed. Process models are used to quantitatively identify, isolate, and measure the different sources of error in a reconstruction. Modeling the way in which a proxy registers climate information formally (Evans et al., 2013) opens new opportunities for integration with other proxies, hypothesis testing, and for assimilation into more complex ecosystem models. Tipton et al. 2016, for example, developed a Bayesian hierarchical statistical model of tree-ring growth to

jointly reconstruct temperature and precipitation at a site in the Hudson Valley, NY. Their multi-scale model allows tree growth to vary by species and to respond to both temperature and precipitation variations in climate through time at different temporal scales. Quantification of uncertainty at this level of rigor is only possible when calibration data are available for the appropriate spatial and temporal domain of a reconstruction. Even here, however, decisions still remain as to how to assess the accuracy of the reconstructions.

## 3.3   Reconstructed Temperature Variations in the NE US for the Last 3,000 Years

NE US climate changes during the late Holocene were more subtle than those of the early and mid-Holocene (Shuman and Marsicek, 2016). The reconstructed average mean temperature of the warmest month (MTWA) from the 863 pollen sites in the region (Fig. 1I, Williams et al. 2011) during the late Holocene shows that peak summer warming occurs between 2500 and 3000 years ago, with generally declining temperatures thereafter until the most recent two centuries. The mean annual temperature

anomalies based on pollen records from across North America from Viau et al. 2006 also show a long-term but more subtle decline. Long-term cooling during the past millennium is evident in tree-ring (Esper et al., 2012) and other hemispheric- to global-scale multi-proxy reconstructions (Mann et al., 2009; Ljungqvist, 2010). Many of the reconstructions tend to be biased towards summer season temperatures because they rely heavily on tree rings (maximum latewood density), isotopic analyses from speleothems, or lake sediments that largely reflect summer processes (Hu et al., 2001). A cooling trend is also evident in

the SST records (Fig. 5A, B) (Keigwin et al., 2003; Sachs, 2007) and in the transient climate simulations of (Liu et al., 2009) and (Otto-Bliesner et al., 2015) (Fig. 5G, J). The reconstructions by (Viau et al., 2006) and (Williams et al., 2011) draw largely from the same data, but employ slightly different reconstruction methods and reflect very different spatial domains.

Declining temperatures during the past three millennia in the NE US are consistent with the continuation of broader neoglacial cooling since the early Holocene (Gajewski, 1987), due primarily to decreasing summer insolation. Regional cool-



ing is evidenced by changes in forest composition, including increases in boreal forest components such as spruce (*Picea*) and fir (*Abies*) (Davis, 1980; Shuman et al., 2004; Oswald et al., 2007). The pollen-based temperature reconstruction shows the cooling trend clearly (Williams et al., 2011) (Fig. 5I).

Centennial-scale temperature variations in the SST, tree ring, and multi-proxy records (Esper et al., 2002; Trouet et al., 2013; Ljungqvist, 2010) lack strong coherence, which may partly reflect geographic climate variability, but several features are identifiable in multiple records. For example, most of the reconstructions show a peak in warmth at the beginning of the MCA about 1000 years ago, particularly the tree-ring, multi-proxy, and pollen data (Fig. 5C, 5D, 5G-I). Hemispheric temperatures during the MCA were slightly lower than modern temperatures (Ahmed et al., 2013), and this appears true in the Mann et al. 2009, and some of the pollen-based reconstructions (Fig. 5C-D, 5H-I), but these pollen-based reconstructions do not account for increasing vegetation disturbance towards present. Warming at the onset of the MCA generally gives way to continued cooling, and relatively low temperatures are recorded in diverse proxies during the LIA. Aside from the Viau et al. (2006) pollen-based reconstruction (Fig. 5H), temperatures generally cool again from the beginning to the end of the LIA.

Recent warming is evident in most of the temperature reconstructions and also in the climate simulations, generally beginning one to two centuries ago and increasing steadily towards present. Historical temperature data become abundant at this time and offer more detailed insight into patterns and trends (Fig. 6). Consistent with the National Climate Assessments from the Midwest and Northeast (Kunkel et al., 2013), EOF analysis of the historic weather station data from the GHCN indicate a general warming trend in both the eastern and western parts of the NE. The warming trend is not necessarily consistent across seasons, as increased greenhouse gases are causing nights to warm faster than days, and this can reduce rather than increase tree stress. There is a strong west-to-east gradient however, in temperature variations over the past century (Fig. 6A, B). The EOFs account for 16% and 8% respectively (the next are 5, 4 and 3% respectively). The west (northwest especially) showed an early- to mid-century maximum in temperatures and high variability (green line in 6C), while the Atlantic margin experienced strong warming throughout the century (Fig. 6C, brown line). The second EOF (Fig. 6B, D) shows a northwest to northeast temperature gradient, which is largely inverse to the first EOF (Fig. 6A, C). In the secondary EOF pattern, higher variability is evident in the warming trend of annual temperatures from the northwest (Fig. 6D, brown line) than the same pattern along the Atlantic seaboard (Fig. 6C, brown line). Likewise, the early- to mid-century warm period (Fig. 6C, D, green lines) is less variable along the Atlantic seaboard (Fig. 6D), and smaller in magnitude than the same pattern in the northwest (Fig. 6C).

Given the regional NE US paleo data, it is difficult to accurately assess how current temperature increases compare with the late Holocene observations, especially during the MCA, for example. The recent reversal of the long-term cooling trend is very clear, however. Further analyses of pollen and tree-ring data, along with continuing development of new regional paleo temperature reconstructions are needed to better contextualize current warming, and to examine the spatial expression of regional climate variations during the MCA and LIA. The relationship between seasonal variations and long-term trends also remains unclear, but could be potentially be revealed with further paleo data analyses. Additional studies may also help determine synchrony and asynchrony of temperatures between the NE US and other regions.



### 3.4 Reconstructed Hydroclimate Variations in the NE US for the Last 3,000 Years

The variety of paleohydrological records available for the NE indicate a long-term increase in effective moisture as well as several extreme, short-lived hydroclimatic events, including both droughts and pluvials (anomalously wet periods) (Fig. 7). The most widespread drought of the late Holocene in the NE US occurred between 550 and 750 CE (1400-1200 cal yr BP), as

evidenced by decreased lake levels across many sites, lowered bog water levels, and reduced tree-ring widths. Sluice Pond in northeastern Massachusetts (not shown) also indicates drought 2000 to 1300 years ago, which overlaps in time with this event (Hubeny et al., 2015).

    Tree-ring data show variability in the PDSI with annual resolution during the past 1400 years, and a trend towards increasing moisture in the east (Fig. 7A). A severe dry period is evident ca. 1300-700 cal yr BP (650-1250 CE), and a widespread event (in

both series) is evident ca. 1150 cal yr BP (800 CE). The most intense drought appears in the North Atlantic coastal sites during the MCA. Similar to the eastern PDSI data, the varve thickness data (Fig. 7B), and the eastern bog records (Fig. 7C) show increasing wetness or a reduction in the number of drought events during the past 9000 years (Fig. 7B). Hubeny et al. (2011) show that dry periods in the varve record correspond to several well-documented events including the MCA, the Dust Bowl in the early 20th century, and the 1960s drought. The varve-based index is not strongly correlated with the tree-ring-based PDSI

reconstruction ($R^2$=0.13 on the smoothed data) however, and we found no correlation on an annual basis ($R^2$=0).

    The bog data extending back to 1000 BCE (2950 cal yr BP) suggest several early periods of increased drought frequency, such as around 550 BCE (2500 cal yr BP) and 100 CE (1850 cal yr BP), but these events are not evident in the lake level records (Fig. 7F). The SVR record (Fig. 7D) does indicate relatively dry conditions/increased drought around 100 CE, however. Between about 400 and 900 CE (1550-1050 cal yr BP), the lake and tree-ring records show abrupt shifts in hydroclimate. Many

of the lake-level reconstructions first shift wetter conditions but subsequently shift to drier conditions ca. 700 CE (1250 cal yr BP). Progressively wetter conditions follow in the varve, lake-level records, and eastern PDSI record (Fig. 7A-B, F). The White Lake, New Jersey IRM records (Fig. 7E) do not capture long-term trends, but do show two drought events, ca. 3000 years ago and again from 400-900 CE (1550 to 1050 cal yr BP) (Li et al., 2007). The drought around 400 CE (1550 cal yr BP) registers in both cores, with the core from the shallower part of the lake showing low water levels from about 400 and 600

CE (1550-1350 cal yr BP), and the core from the deeper part of the lake showing a moderate increase in magnetism between 400-600 CE (1550-1350 cal yr BP) followed by a larger increase (i.e., strong reduction in lake levels) from 600 to 1000 CE (1350-1050 cal yr BP).

    The lake-level reconstructions from Davis and New Long Ponds generally show progressively wetter conditions during the past 3000 years (Fig. 7F). Deep Pond shows a slightly different pattern, with a rapid decrease in lake levels ca. 3000 years

ago followed by more variable but generally increasing lake levels towards present. The local maxima in lake levels ca. 400 CE (1550 cal yr BP), followed by local minima between 500 and 800 CE (1450-1150 cal yr BP) appears to reflect widespread shifts in moisture availability as these events are also evident in the bog and IRM records in the centuries prior to the MCA. The testate amoeba records (Fig. 7C) shift from infrequent to more frequent drought from ca. 400-1000 CE (1550-950 cal yr BP), and lake levels (Fig. 7F) shift from high to low, then back again, especially at Deep Lake. The PDSI records (Fig. 7A)



also show large oscillations during this period. It is unclear whether increasing drought ca. 1400 years ago occurred against a backdrop of warming or cooling temperatures (Fig. 5), but rapid shifts between wet and dry intervals alone may help explain a sharp rise in chestnut and hemlock, coupled with a large increase in charcoal influx that was observed in paleodata from Massachusetts at this time (Foster et al., 2002; Parshall et al., 2003). In the most recent century, an acceleration of the gradual

wetting trend is observed in PDSI records, the varve index, and the lake level reconstruction from Whitehead Lake. The SVR data show increasing drought during recent centuries, but human influences are known to have increased near the bog in recent centuries, which may have disturbed water levels.

Spatial structure in the NADA data from 1700 to 2005 CE shows three primary patterns (Fig. 8), although none of these explain substantial variance. The strongest pattern (13% of the variance explained) shows drought centered in the Midwest,

similar to the historical drought of 1931-1940 (the "Dust Bowl"). The second EOF (9% of the variance) has a pattern similar to the southwest drought of 1950-1957. A third EOF (explaining 8% of the variance) reflects widespread drought in the southeast, similar to the 1889-1896 event (Herweijer et al., 2005) and also to the 1960s drought. The low variance explained by any one particular EOF means that drought patterns have been heterogeneous in the recent past. The pattern reflected in the third EOF, however, shows that widespread drought in the east is not uncommon, consistent with the coherent drought in the paleo

hydroclimatic records about 1200 years ago (Fig. 7).

Short-lived disturbances such as storms can have dramatic impacts on ecosystems in the NE US (Besonen et al., 2008; Donnelly et al., 2001). Erosion patterns inferred from particle size variations in six northern New England lakes suggest an increase in extreme precipitation events during recent millennia, for example Parris et al. (2010), but Noren et al. tell a slightly different story with the same approach for New Hampshire. A synthesis of all New England storms of at least tropical storm

impact shows no long-term trend in either hurricanes or all tropical cyclones (Fig. 9). Storm activity was high in the 1950s, however, and low in the early 20th century and mid-late 18th century. The storminess during the 1950s may be part of a broader pattern of Atlantic hurricane activity related to the AMO (e.g. Goldenberg et al., 2001). About 0-2 hurricanes occurred per year since 1743, and major hurricanes (Category 3, 4 or 5 on the Saffir Simpson scale) are very rare. Relative stability in hurricane occurrence during the past 260 years highlights the importance of unique daily synoptic patterns conducive for

tracking storms towards New England, and suggests that forcings like solar activity and greenhouse gases have had limited effect on overall numbers in New England. Storm activity does not appear anomalous during the end of the LIA. Some past storms with hurricane-force winds are likely similar to those in recent memory, such as post-hurricane Sandy in 2012, the Perfect Storm in 1991, and the Halloween Nor'easter of 2011.

## 4 Conclusions

With the exception of fossil pollen records, there are a limited number of northeastern US (NE US) paleoclimate archives that can provide constraints on its temperature and hydroclimate history. The regional paleoclimate data that do exist show that the current warming and wetting trend reflects a reversal of millennial-scale cooling and wetting trends prior to the 1800s. Although paleo temperature data in the region are very limited, pollen data, coastal SST marine records, and broader-scale





multi-proxy networks indicate regional long-term pre-industrial cooling. Evidence for a trend towards wetter conditions in the past 3000 years is strong in the lake-level data, and for the past millennium. Regional testate amoeba records cannot be used to detect low frequency hydroclimate trends however, and the divergence between these data and the lake-level data remain unresolved. Taken together, the available paleo evidence suggests that the climate was both warmer and drier during the MCA

5    than during the LIA.

Multiple types of paleo data – from decreased lake and lowered bog water levels to reduced tree-ring widths – suggest that the most widespread drought of the late Holocene in the NE US occurred between 550 and 750 CE (1400-1200 cal yr BP). No trend is evident in storm frequency in New England, but increasing rates of change in both growing-season temperature and moisture will likely have significant impacts on vegetation and disturbance regimes. New calibrated paleoclimate records will

10   help constrain spatiotemporal variations in temperature and precipitation in the NE.

*Author contributions.* JRM, BS, SG, CN, and NP designed the study, RB, EC, AD-K, CM, JN, NP, BS, and ZY provided data, JRM, SG, CN, and PJB analyzed the data, JRM, NP, and SG wrote the initial manuscript draft, and all authors contributed to writing and editing

*Acknowledgements.* We thank Matthew Montanaro for assistance with data analysis. This work was supported by NSF grant EF-1065732 to SJ, JW, and JM and by NSF grant EF-1241870 to JRM, NP, RB, SJ, JM, DJPM, and JW.



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





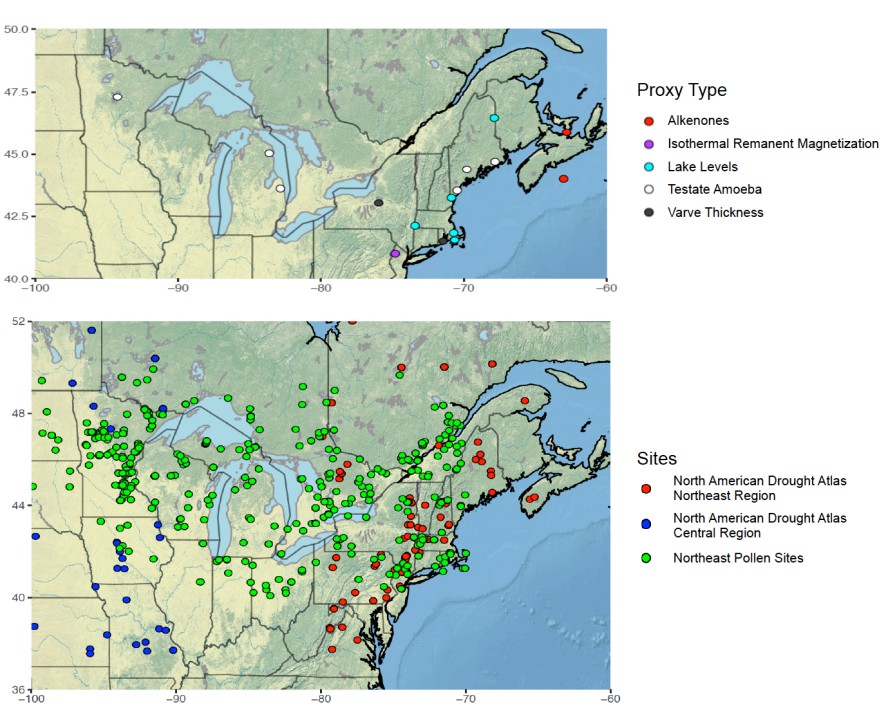

**Figure 1.** Locations of sites with paleoclimate data used in this paper and that span at least two centuries, reconstructed from various sources in the northeastern US.



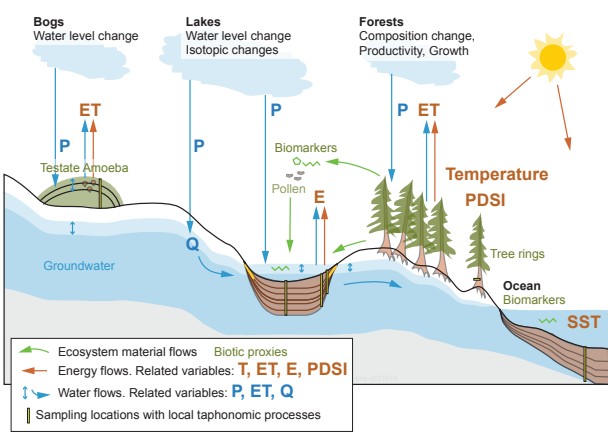

**Figure 2.** Natural archives such as sediments, bogs, and trees rings capture signatures of energy, water, and other material flows between the biosphere, surface hydrosphere, and atmosphere. Paleoclimate proxies are the various biological, geochemical, and physical characteristics of these archives that are influenced by climatic variations. Ombrotrophic bogs are perched above the water table and thus they and their testate amoebae assemblages are not affected by ground water flows (Q). Rather, the composition of testate amoebae assemblages within bogs is governed by the differential sensitivity of testate amoebae species to the height of the water table within the bog, governed by the balance between precipitation (P) and evapotranspiration (ET), which is in turn affected by temperature (T). Lake level records and diatom-based records of past salinity integrate decadal to century-scale T, P, and ET variations. Tree ring growth responds to seasonal T and P variations, and are often used to reconstruct the Palmer Drought Severity Index (PDSI). Biomarker, isotopic, and pollen signals recorded in bog and lake archives capture decadal or longer-term processes due to processes that affect particle transportation and deposition. The proxies found in marine archives are mostly produced by marine organisms (e.g. biomarkers) but some such as pollen are terrestrially sourced.





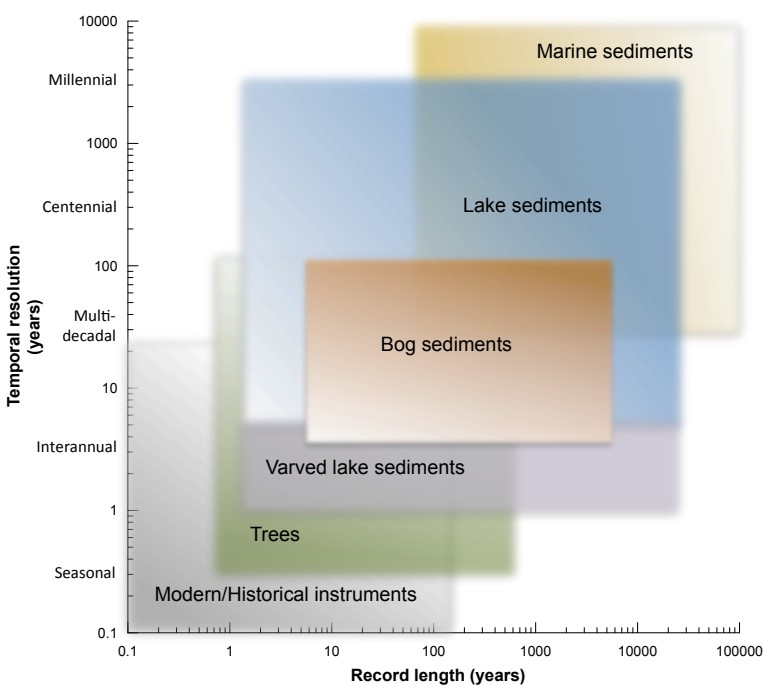

**Figure 3.** Temporal resolution and record lengths for paleoclimate records commonly used in the NE US.



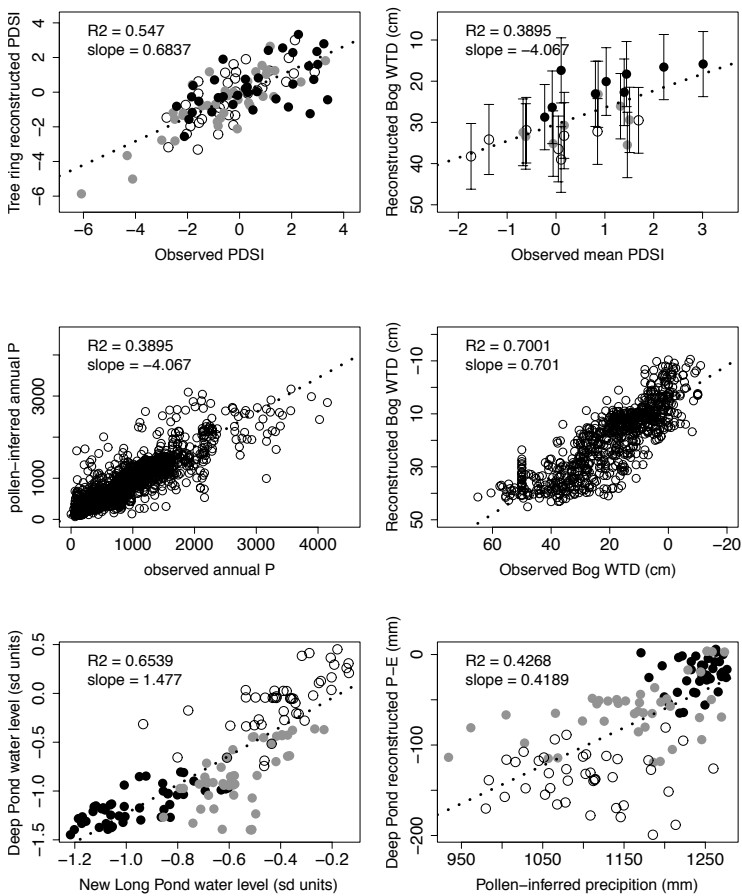

**Figure 4.** Calibration, internal validation, and multi-proxy comparison data for tree rings, pollen, lake levels, and bogs. Top row: instrumental (observed) versus reconstructed values correlated in time. Left: NY PDSI for 1895-2000; right: Hole Bog, MN water-table depth versus instrumental PDSI for 1895-2003 (which uses the mean PDSI for each of the sample intervals) (Booth, 2008). Middle row: observed versus predicted paleohydrological variables, based on proxies found in lake and bog archives, and inferred using space-for-time paleoclimatic transfer functions and validated using cross-validation. Left: pollen data from (Marsicek et al., 2013); right: testate amoebae data from across North America (Booth, 2008). Bottom row: inter-site and cross-proxy comparisons using lake-level and fossil pollen records from the NE US. Note that these comparisons include reconstruction error as well as temporal error because the two datasets in each case come from different sets of cores. Left: New Long versus Deep Pond water elevations from southeastern MA for the past 7000 yrs (Newby et al., 2014); Right: Deep Pond lake-level derived P-E reconstruction versus the mean annual precipitation reconstruction based on the mean of multiple records in the region (Newby et al., 2014). For all plots, black-filled dots are the oldest one third, gray are the middle third, and unfilled are the youngest third.



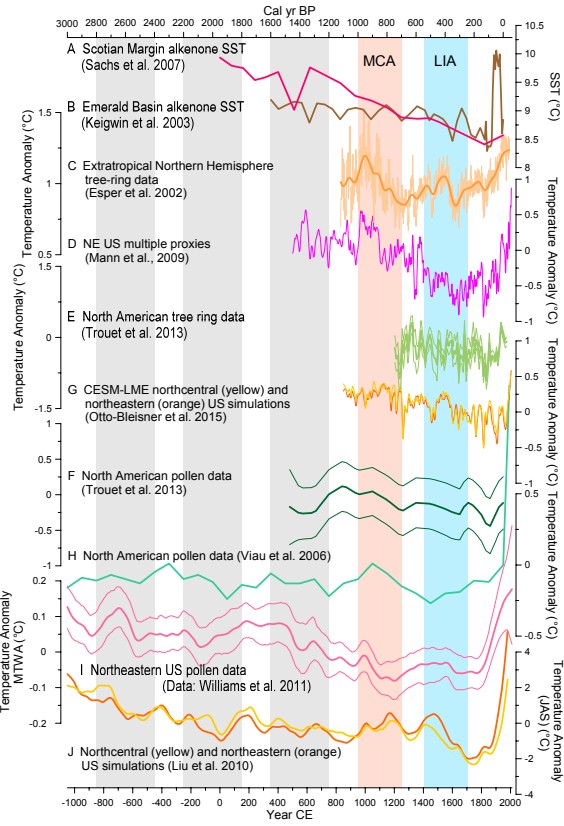

**Figure 5.** Temperature reconstructions for the last 3000 years for the NE US and adjacent regions, sourced from diverse paleoclimatic proxies and archives. A) Sea Surface Temperature (SST) reconstructions based on $U^K_{37}$ in marine sediments (Sachs, 2007); B) SST based on alkenone unsaturation in marine sediments (Keigwin et al., 2003); C) tree-ring widths from the northern hemisphere extratropics (Esper et al., 2002); D) Northeast US grid cells areally averaged (from 37.5 to 47.5 N latitude and 67.5 to 77.5 E longitude) from multiple proxies (Mann et al., 2009); E) tree-ring widths from temperate North America (Trouet et al., 2013); F) North-central (39 to 46 N latitude and 87.5 to 97.5 E longitude; yellow) and Northeast (39 to 46 N latitude and 67.5 to 77.5 E longitude; orange) simulations from the CESM-LME (Otto-Bliesner et al., 2015); G) North American pollen data (Trouet et al., 2013); H) North American pollen data Viau et al. (2006); I) Northeastern pollen data (Williams et al., 2011); J) North-central (yellow) and Northeastern (orange) temperature simulations from the CCSM3 SynTrace experiment (Liu et al., 2009). Vertical gray bands are only provided as a visual aid to identify the timing of features in the records. Pink and blue vertical bars mark the well-known MCA and LIA intervals.





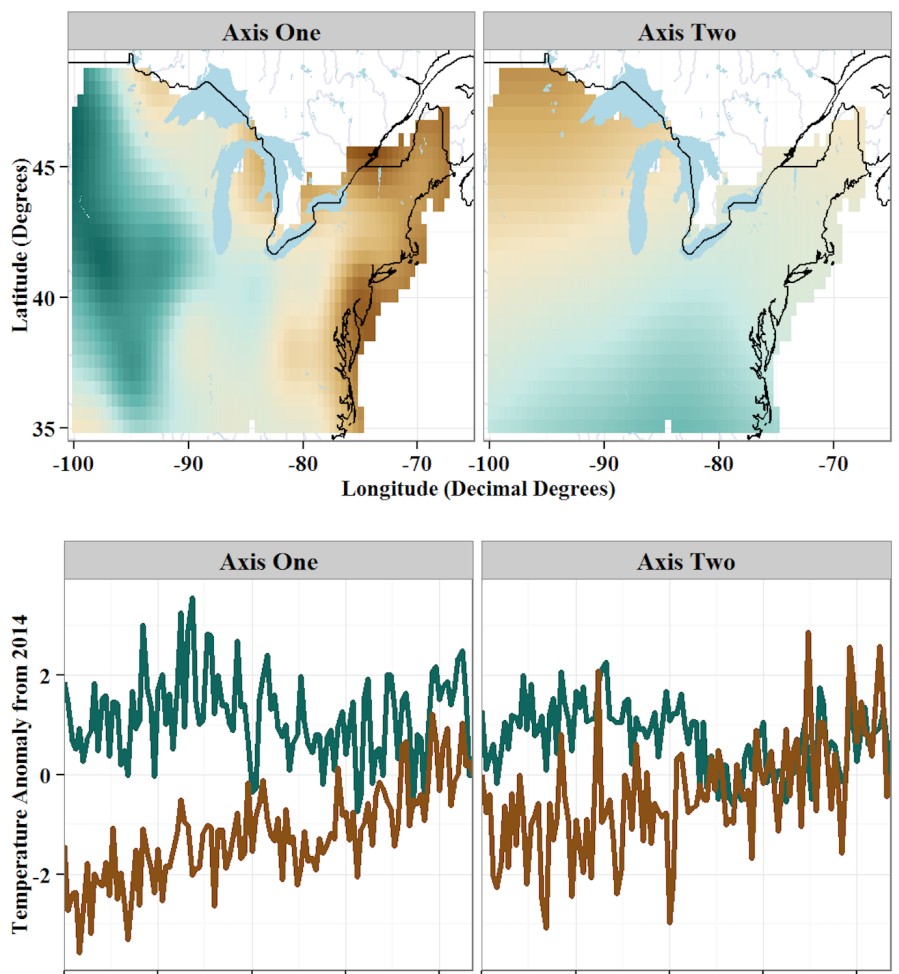

**Figure 6.** Broad-scale spatiotemporal patterns in the historical mean annual temperature anomalies from 1895 - 2010 from an empirical orthogonal function (EOF) analysis of data from the Global Historical Climatological Network for the NE US. A) The first EOF (16% of the variance) shows that the primary temperature changes across the NE US during the last century were a west-east (longitudinal) difference, where the west (northwest especially) showed an early- to mid-century maximum and high variability (green line in c), while the Atlantic margin experienced strong warming throughout the century (brown line in c). The second EOF (map: b, time series: d, accounting for 8% of the variance) effectively reverses the first EOF, but indicates that the broad warming trend in the northwest shows much higher variability (brown line in d) than the same pattern along the Atlantic seaboard (brown line in c), while the early-mid century warming trend is less variable along the Atlantic seaboard, and smaller in magnitude (green line in d) than the same pattern in the northwest (green line in c).



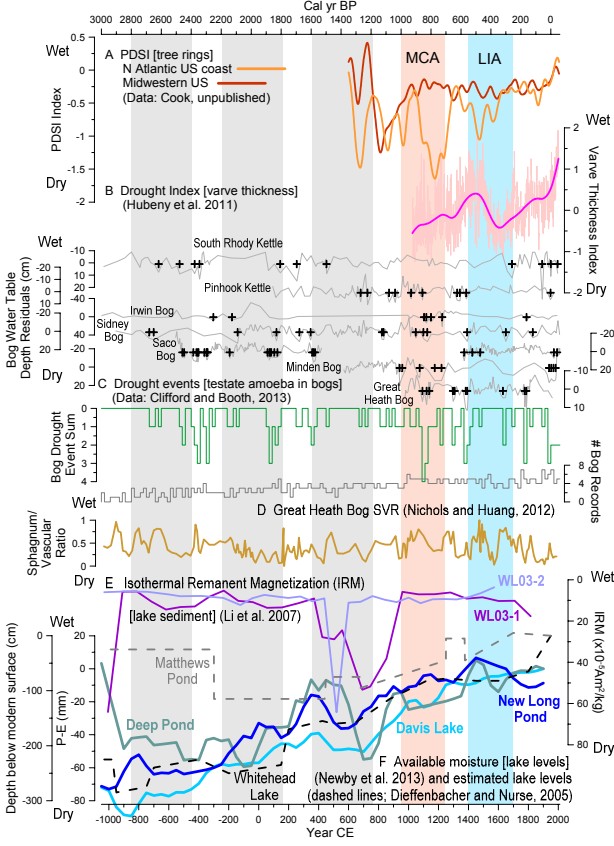

**Figure 7.** Hydroclimatic reconstructions for the last 3000 years for the NE US and adjacent regions, sourced from diverse paleoclimatic proxies and archives. A) tree-ring based reconstructions of PDSI (Cook et al., 2010)); B) composite index of two varve thickness records from Green Lake, NY, and the Pettaquamscutt River Estuary, RI (Hubeny et al., 2011); C) drought events and event summary based on water table reconstructions from testate amoeba (Sidney, Irwin Smith, Great Heath, Saco, and Minden bogs, and Pinhook and South Rhody kettle lakes)(Booth and Jackson, 2003; Booth et al., 2006, 2012; Clifford and Booth, 2013); D) ratio of *Sphagnum* to vascular plants from the Great Heath bog, ME (Nichols and Huang, 2012); E) drought indicator based on Isothermal Remanent Magnetization at White Lake, NJ (Li et al., 2007); and F) lake level reconstructions from Deep Pond, Davis Lake, New Long Pond (Marsicek et al., 2013; Newby et al., 2014), and lake-level change estimates from Matthews Pond and Whitehead Lake, ME (Dieffenbacher-Krall and Nurse, 2005). Vertical gray bands are only provided as a visual aid to identify the timing of features in the records. Pink and blue vertical bars mark the well-known MCA and LIA intervals.





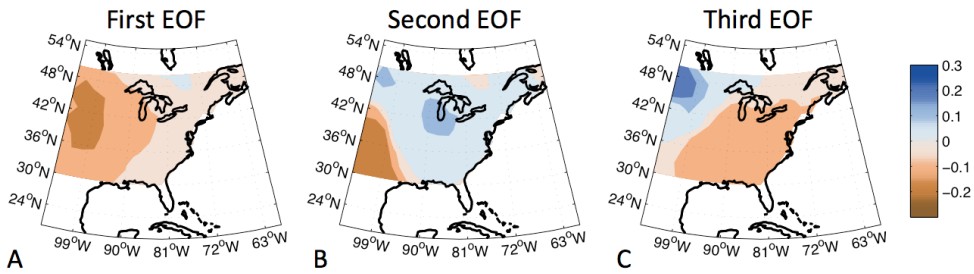

**Figure 8.** NADA spatial structure and signals. First three EOFs for the 2.5°NADA. Calculated for 1700-2005. Variance explained: 13%, 9%, and 8%. Colors indicate areas with opposite loading patterns for relatively wet versus dry conditions.

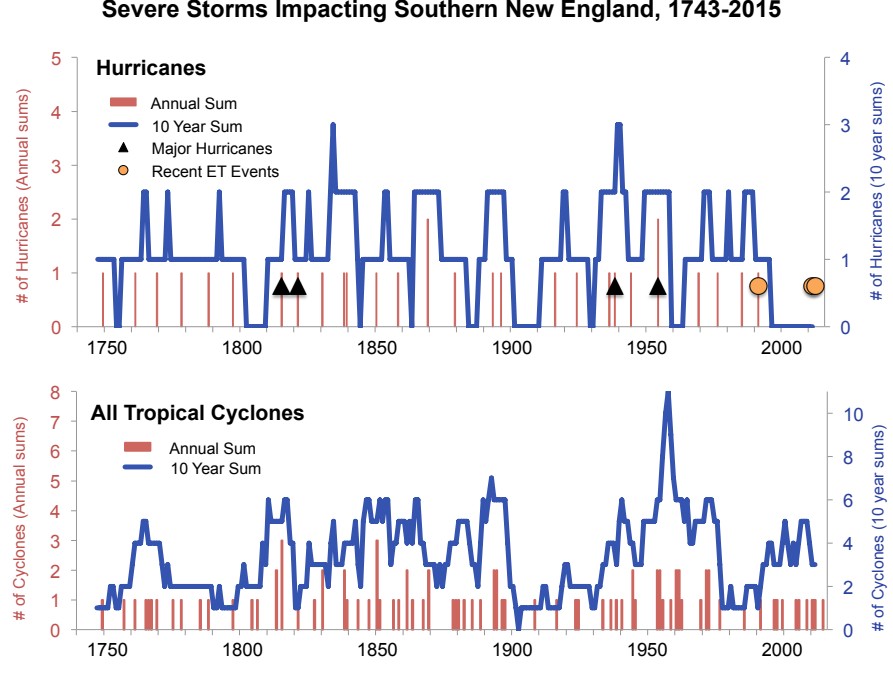

**Figure 9.** Annual number and decadal sums for a) reconstructed hurricanes and b) tropical cyclones and hurricanes in southern New England from 1743 to 2015 CE. Major hurricanes (i.e. Saffir Simpson Category 3, 4 or 5) are shown as triangles. Selected recent extra tropical storms (the Perfect Storm in 1991, the Halloween Nor'easter of 2011, and post-hurricane Sandy in 2012) are shown as circles. Methodology for the storm reconstructions follow those outlined in Mock (2008).