# Peer review of "Climatic history of the northeastern United States during the past 3000 years"

_Climate of the Past, 2016_

## Referee Comment (RC1) · Anonymous Referee #1 · 9 Dec 2016

Major comments:

This study conducted a systematic review, assessment, and comparison paleotemperature and paleohydrological proxies from the NE US for the last 3000 years. Based on the comparison and cross-check between different paleotemperature and paleohydroclimate reconstructions from the NE US, the main conclusion that the current warming and wetting trend reflects a reversal of millennial-scale cooling and wetting trends prior to the 1800s was drawn. There are several parts which should be improved. In particular, Figure 5, 6, 7 and related discussion in the manuscript that draw the main conclusion of the study should be revised and polished for a better understanding for readers. In addition, seasonality should be considered when different reconstructions are compared. For example, several records may reconstruct annual temperature, while others reconstructed summer or mean temperature of the warmest month temperature. Detailed reconstructed target for each record should be added.

Specific comments:

Figure 4: This figure has 6 panels, adding the a, b, c, d, e, f for each panel is helpful for a smooth reading. Now the order of figure is confusing.

Page15 line 9: Figure 5A, should be Figure 4e (New Long versus Deep Pond water elevations from southeastern MA for the past 7000 yrs).

Page15 line 12: Figure 5B should be Figure 4f (Deep Pond lake-level derived P-E reconstruction versus the mean annual precipitation reconstruction based on the mean of multiple records in the region).

Page15 line 19: Figure 5C should be Figure 4c (observed versus predicted paleohydrological variables, based on proxies found in lake and bog archives, and inferred using space-for-time paleoclimatic transfer functions and validated using cross-validation. Left: pollen data from (Marsicek et al., 2013)).

Page15 line 31: Figure 5D should be Figure 4d (observed versus predicted paleohydrological variables, based on proxies found in lake and bog archives, and inferred using space-for-time paleoclimatic transfer functions and validated using cross-validation. right: testate amoebae data from across North America (Booth, 2008)).

Page16 line 4: Figure 5E should be Figure 4A (instrumental (observed) versus reconstructed values correlated in time. Left: NY PDSI for 1895-2000).

Page 16 line 23: Figure 1I should be Figure 1 or Figure 5I.

Page 16 line 24-26: The mean annual temperature anomalies based on pollen records from across North America from Viau et al. 2006 (Figure 5H) also show a long-term but more subtle decline.

Page 16 line 31-32: add more explanation on the reason why long-term difference between reconstruction by Williams et al. 2011 (Figure 5I) and Viau et al., 2006 (Figure

5H).

Figure 5: the x axis should not contain "0" for CE. Same for Figure 7.

Figure 6: add the loading legend. Which color indicates positive loading? Adding the a, b, c, d to each panel.

Page 17 line 19-26. The sentence is not clear. Does brown line (Figure 6c) means temporal variations of EOF1?

Page 18 line 12: during the past 9000 years should be 900 years. Figure 7c does not show clear long-term trend. Maybe it does not preserve low frequency signal so much.

Page 18 line 14-15: Could you give explanation why there is no correlation between tree-ring based PDSI and varve-based index? Both tree ring and varve records could be calibrated with instrumental data, but they are not correlated. Please give more information on reconstructed PDSI, seasonal PDSI? Or annual? Same as varve-based reconstruction.

---

## Referee Comment (RC2) · Anonymous Referee #2 · 19 Mar 2017

The authors attempted to give a complete overview of climatic history of the northeastern United States in the context of the past 3000 years. It is crucial and helpful to assess current climate conditions in the sparsely-distributed study region. They tried to combine different proxy records with high-resolution and lower-resolutions, including lake, bog and varve sediments, pollen and marine records as well as tree rings. They firstly give an overview on each of proxy archives in terms of basic principles recording climate signals, and their strengths and weaknesses separately, and then reconstructed temperature and hydroclimate variations over the late Holocene by comparing different proxy records. They argued that the NE US witnessed a regional long-term pre-industrial cooling but reversed after the 1800s. As for hydroclimate variations, they identified the most widespread drought in the NE US occurring during the period 550-750 AD. However, the divergence between Regional testate amoeba records and the

lake-level data remain a big question and new paleoclimate records are necessary to resolve the differences. Generally, the paper is well-written. Their efforts have added new contribution to the knowledge of regional climate variability and to site selection of new data collection. The detailed description on each of proxy records is informative and an excellent addition. This work is worthy of publication in the Climate of the Past. However, there are some concerns should be clarified or explained before it is ready to go.

Comments:

My major concern is that the sections 3.3 and 3.4 seem to be a little bit confusing. I suggest that they set out a priority of different proxy records for organizing their discussion and defining the special climate events. For example, tree rings is ranked the first priority, varve thickness second, pollen third, and so on. Specific comments are as below.

1) Page 3, line 29, I am confused with 'bogs with groundwater-sourced inputs'. Do you mean that the bog is influenced by groundwater? 2) Page 10, line 13, please rephrase this sentence. 3) Page 16, lines 18-21, it is important to clarify why the two reconstructions have very different trends considering both used the same dataset. 4) Page 18, lines 11-12 are confusing and need clarification. It is not the case by visual inspection. 5) Page 20, line 6: Visual inspection did not show that bog water levels were lower at that time. 6) Figure 1: 'Proxy type' should be proxy type and sites? It is unclear that each of the pollen sites hosts a long-term climate history? 7) Figure 2: What do you mean by "E" in the Figure 2? It needs full name. The same is for Q, and others. Please check on the figure 2 to be sure all the descriptions are correct and complete. 8) Figure 3: it is confusing and somewhat misleading why the temporal resolution (vertical axis) of trees varies from seasonal to 100 years. Please clarify and confirm it. The same question is for modern/Historical instruments. 9) In Figure 4, bottom row, right panel: correct 'precipition' as precipitation. What do you mean by 'NY, MN, and others'? Please provide the full name, and the compared common periods

are also needed for clarification. 10) Figure 5: It would be more logical to arrange the simulation curves together. Anyway, the curves in Figure 5 need to be re-arranged. The pollen data F, H and I are very different from each other. Which is more reliable? The difference is caused due to regional temperature differences as identified in Figure 6 or different methods or dating uncertainties? More discussion is needed for clarification, rather than just listing them together. I don't know why the G and F figures were arranged reversely. Did the North central and northeastern US simulations correspond to the west and the Atlantic margin in Figure 6 respectively? Liu et al., 2010 did not match with Liu et al. 2009 in the reference list. The Liu et al. 2009 simulation represents a long-term decreasing trend which is similar to most of reconstructions, but it is different from reconstructions in term of on medium-frequency domains. Also, I miss some discussions on driving mechanism in the study region considering that they used the GCM simulation results. To my knowledge, Esper et al. (2012) identified an orbital forcing in tree-ring data in high NH latitudes (to the north of 65 degrees north latitude), but the forcing weakens quickly towards middle- and lower-latitudes. The study region in this manuscript is located in middle latitudes. Perhaps the orbital forcing is not strong or non-existing. Anyway, more discussion is helpful for mechanism hinted for the last 3000 years. Figure 7: The authors presented spatiotemporal patterns in the historical mean annual temperature anomalies from 1895 to 2010 as shown in Figure 6. Is it possible to organize or discuss the temperature variations over the last 3000 years separately for the east and west part of the study region, considering that the two parts of the study region represent very different temporal variations in temperature and it is possible that the same situation occurred in the past too. On the other hand, the Figure 6 represents the variations in mean annual temperature. It is recommended to present similar figures or results in the text because most of proxy records are sensitive to summer temperatures. The same suggestion is applied to moisture variations. Maybe a figure similar to the Figure 6 is helpful to guide the authors to organize the different archives based on the sub-regions in the study area. I guess that the moisture variations might have differed more regionally than temperature. 11) Figure 7: What

do theymean by the '+' symbol? Please clarify it in the Figure legend. How about the dating uncertainties for lake or bog sediment records? It is crucial for comparison with other high-resolution data. It is possible that the dating accuracy affects poor correspondence among the proxies. An additional table outlining all the details including dating material, sampling resolution, temporal resolution, dating methods and so on is useful. As a result, a separate discussion paragraph is needed. What is the full name for NY, RI, ME and others? All the testate amoeba data are from the same region, west or eastern region? From Figure 1, the related data distribute widely. Are they arranged regionally or not? Can they provide the simulation output results for moisture variations over the past 3000 years from GCM modeling? If so, it would be a very helpful addition.

---

## Author Comment (AC1) · 24 Apr 2017

Referee #1 requests two key areas in need of improvement in the paper: 1) revisions to Figures 5, 6, and 7 to facilitate understanding and interpretation of the paleoclimate data, and 2) additional information and explanation of the seasonality of different data sources and how this plays out in the reconstructions. We appreciate the recommendations and agree that both of these issues need to be addressed. We also appreciate the attention to detail on the figures, and can correct each issue identified.

To improve the figures (5, 6, and 7) we will correct the axes and labels, add the loading legend, and reorganize the time series so that they are grouped in a more intuitive fashion, with modeled series together at the bottom. This will allow the interpretation to flow more easily and clearly from the data. To address the second item, we will add

a short explanation of the seasonality of each proxy in their descriptive sections, and then tie these to the discussion of the features and trends in the multiproxy figures. For example, we will clarify that the PDSI reconstruction is calculated for June, July, and August, so it is a summer reconstruction, and therefore may reflect different dimensions of drought than those observed in the bog data.

The bog data presented here most closely reflect the length and severity of the summer moisture deficit, which is usually related to summer precipitation (Charman 2007, Booth 2010). There can be important differences in seasonal sensitivities however across both space and time, and the relative importance of precipitation versus temperature may also vary among sites (Booth, 2010). Thus in some cases, changes in water table depth may be more closely correlated with summer PDSI, while elsewhere it may reflect annual PDSI more closely. Still other bogs may respond to temperature more than precipitation variations, which may be correlated with each other and thus produce effects that are very difficult to disentangle. Overall then, it is not surprising that a summary of inferred drought events from many different bog records across a broad region do not closely parallel tree-ring-based PDSI that reflect a narrower set of physical and biological processes, and differing resolutions and chronological uncertainties makes these comparisons more uncertain. It should be noted, however, that North American bog records are correlated with local PDSI data (e.g., Booth 2010) and similar correlations with temperature and precipitation have been found at European sites (Charman et. al 2004, 2012; Charman 2007; Schooling et al. 2005).

Figure 4: This figure has 6 panels, adding the a, b, c, d, e, f for each panel is helpful for a smooth reading. Now the order of figure is confusing.

We will add panel labels and change the order of the panels to be more intuitive, moving from the most direct data comparisons (with the least uncertainty) to the more indirect comparisons (with more uncertainty). The top row will show observed versus reconstructed PDSI from tree rings (left) and observed versus reconstructed water table depths (WTDs) from bog data (right). The middle panel will show observed versus

reconstructed annual precipitation from pollen (left) and observed PDSI versus reconstructed bog WTDs (right). The bottom panel will show reconstructed precipitation from pollen versus reconstructed P-E from lake level data (left) and reconstructed lake levels from two different lakes (right).

Page15 line 9: Figure 5A, should be Figure 4e (New Long versus Deep Pond water elevations from southeastern MA for the past 7000 yrs).

Agreed; we will fix this.

Page15 line 12: Figure 5B should be Figure 4f (Deep Pond lake-level derived P-E reconstruction versus the mean annual precipitation reconstruction based on the mean of multiple records in the region). Agreed; we will fix this.

Page15 line 19: Figure 5C should be Figure 4c (observed versus predicted paleohydrological variables, based on proxies found in lake and bog archives, and inferred using space-for-time paleoclimatic transfer functions and validated using cross-validation. Left: pollen data from (Marsicek et al., 2013)).

Agreed; we will fix this.

Page15 line 31: Figure 5D should be Figure 4d (observed versus predicted paleohydrological variables, based on proxies found in lake and bog archives, and inferred using space-for-time paleoclimatic transfer functions and validated using cross-validation. right: testate amoebae data from across North America (Booth, 2008)). Agreed; we will fix this.

Page16 line 4: Figure 5E should be Figure 4A (instrumental (observed) versus reconstructed values correlated in time. Left: NY PDSI for 1895-2000).

Agreed; we will fix this.

Page 16 line 23: Figure 1I should be Figure 1 or Figure 5I. It is Figure 1; we will fix this.

Page 16 line 24-26: The mean annual temperature anomalies based on pollen records

from across North America from Viau et al. 2006 (Figure 5H) also show a long-term but more subtle decline.

Agreed; we will adjust text to reflect this.

Page 16 line 31-32: add more explanation on the reason why long-term difference between reconstruction by Williams et al. 2011 (Figure 5I) and Viau et al., 2006 (Figure 5H).

The reconstructions by Viau et al. 2006 and Williams et al. 2011 reflect differences in the underlying fossil pollen data, age models, the modern calibration datasets, and the reconstruction method, although the methodological differences are likely the least important factor. Viau et al. 2006 used 752 pollen records and 4590 modern calibration samples. We used 863 pollen records from Williams et al. 2011, which had updated age models based on linear interpolation between age controls (Williams et al. 2004) and 4833 modern calibration samples (Whitmore et al. 2005). Both studies used slightly different versions of the modern analogue technique, which numerically compares the differences in fossil pollen composition of each sediment sample with the composition of each modern sample from a calibration dataset. The environmental characteristics of the most similar modern samples are then averaged and assigned to the target fossil sample (Williams et al. 2008).

Figure 5: the x axis should not contain "0" for CE. Same for Figure 7.

Agreed; we will fix this.

Figure 6: add the loading legend. Which color indicates positive loading? Adding the a, b, c, d to each panel.

The sign of the loadings is arbitrary; the green on the map corresponds to the green temperature anomalies below, and likewise for the brown colors. Axis one scores are larger than Axis two scores, consistent with PCA more generally. We will add the labels to the panels.

Page 17 line 19-26. The sentence is not clear. Does brown line (Figure 6c) means temporal variations of EOF1?

The brown line refers to the negative loadings of EOF 1; we will clarify this sentence.

Page 18 line 12: during the past 9000 years should be 900 years. Figure 7c does not show clear long-term trend. Maybe it does not preserve low frequency signal so much.

Agreed; we will address this in the text.

Page 18 line 14-15: Could you give explanation why there is no correlation between tree-ring based PDSI and varve-based index? Both tree ring and varve records could be calibrated with instrumental data, but they are not correlated. Please give more information on reconstructed PDSI, seasonal PDSI? Or annual? Same as varve-based reconstruction.

The apparent lack of correlation calculated from the PDSI and varved record was due to differences in the spatial domain for the PDSI reconstruction; we have corrected this so that our results are now consistent with those published in Hubeny et al. – in fact the two series are correlated, most strongly on a mean annual basis.

---

## Author Comment (AC2) · 25 Apr 2017

Referee #2 requests a general revision of sections 3.3. and 3.4 to improve the clarity and understanding of the data interpretation. We appreciate this feedback and agree that some restructuring of the text would be helpful. Currently, we describe the oldest features in the temperature and hydroclimate data, and move towards present, instead of walking through each of the proxies in some kind of priority order. We chose this approach because each of the proxies has different strengths and weaknesses, so in fact we would argue that there is not a natural "priority" as to which is best. In addition, a primary objective of the paper is to understand whether the proxies share common features and signals when considering them all together, which is why we use a chronological framework for interpreting the data. However, we appreciate that our approach is not clearly stated at the beginning of these sections, and thus may

be confusing. To address this, we will clarify our approach to interpreting the data in sections 3.3 and 3.4. We also agree that grouping model data together will help reduce confusion in these sections, which we note in the specific comments below.

1) Page 3, line 29, I am confused with 'bogs with groundwater-sourced inputs'. Do you mean that the bog is influenced by groundwater?

Yes, we will clarify to say simply groundwater.

2) Page 10, line 13, please rephrase this sentence.

Thank you – we will remove "Newby and colleagues".

3) Page 16, lines 18-21, it is important to clarify why the two reconstructions have very different trends considering both used the same dataset.

Agreed. We explain this also in our response to Referee #1, reproduced here: The reconstructions by Viau et al. 2006 and Williams et al. 2011 reflect differences in the underlying fossil pollen data, age models, the modern calibration datasets, and the reconstruction method, although the methodological differences are likely the least important factor. Viau et al. 2006 used 752 pollen records and 4590 modern calibration samples. We used 863 pollen records from Williams et al. 2011, which had updated age models based on linear interpolation between age controls (Williams et al. 2004) and 4833 modern calibration samples (Whitmore et al. 2005). Both studies used slightly different versions of the modern analogue technique, which numerically compares the differences in fossil pollen composition of each sediment sample with the composition of each modern sample from a calibration dataset. The environmental characteristics of the most similar modern samples are then averaged and assigned to the target fossil sample (Williams et al. 2008).

4) Page 18, lines 11-12 are confusing and need clarification. It is not the case by visual inspection.

We appreciate this note – this is an error from an earlier iteration of the figure and we

will update the text to reflect the new figure.

5) Page 20, line 6: Visual inspection did not show that bog water levels were lower at that time.

This is also correct and we will fix this text. The bog data show different patterns than the lake level data during this interval.

6) Figure 1: 'Proxy type' should be proxy type and sites? It is unclear that each of the pollen sites hosts a long-term climate history?

Agreed; we will fix this. We are not certain of the second question here. There is a long history of using pollen records to reconstruct climate, which we describe in the proxy section.

7) Figure 2: What do you mean by "E" in the Figure 2? It needs full name. The same is for Q, and others. Please check on the figure 2 to be sure all the descriptions are correct and complete.

This was indeed missing – E stands for "Evaporation."

8) Figure 3: it is confusing and somewhat misleading why the temporal resolution (vertical axis) of trees varies from seasonal to 100 years. Please clarify and confirm it. The same question is for modern/Historical instruments.

We will change the y-axis label to say "Temporal climate variability reflected by data type" to clarify.

9) In Figure 4, bot- tom row, right panel: correct 'precipition' as precipitation. What do you mean by 'NY, MN, and others'? Please provide the full name, and the compared common periods are also needed for clarification.

We will fix these problems.

10) Figure 5: It would be more logical to arrange the simulation curves together. Anyway, the curves in Figure 5 need to be re-arranged. The pollen data F, H and I are very different from each other. Which is more reliable? The difference is caused due to regional temperature differences as identified in Figure 6 or different methods or dating uncertainties? More discussion is needed for clarification, rather than just listing them together. I don't know why the G and F figures were arranged reversely. Did the North central and northeastern US simulations correspond to the west and the Atlantic margin in Figure 6 respectively? Liu et al., 2010 did not match with Liu et al. 2009 in the reference list. The Liu et al. 2009 simulation represents a long-term decreasing trend which is similar to most of reconstructions, but it is different from reconstructions in term of on medium-frequency domains. Also, I miss some discussions on driving mechanism in the study region considering that they used the GCM simulation results. To my knowledge, Esper et al. (2012) identified an orbital forcing in tree-ring data in high NH latitudes (to the north of 65 degrees north latitude), but the forcing weakens quickly towards middle- and lower-latitudes. The study region in this manuscript is located in middle latitudes. Perhaps the orbital forcing is not strong or non-existing. Anyway, more discussion is helpful for mechanism hinted for the last 3000 years.

We agree that Figure 5 can be reorganized and explained in a clearer manner, grouping the model output in particular. The pollen data span different spatial scales, and for our domain the curve in panel I is the most reliable because it is specific to the northeastern U.S., and includes more records than the reconstruction in panel H (as explained above). Regarding the north central and northeastern curves distinguished in the model data (panels G and J), yes, these do reflect areas similar to those highlighted by the EOF analysis in Figure 6. We can explain this more clearly in the text, and also correct the Liu et al. citation on the figure (it should be Liu et al. 2009). The mechanism behind the long-term decline in northern hemisphere temperatures observed in the pollen data and in the summer temperature simulations is orbital forcing, declining northern hemisphere summer insolation in particular (Liu et al., 2014). The trend is also consistent with a variety of broader paleoclimate and ecological data (e.g., alkenones and charcoal records) from the late Holocene (Marcott et al., 2013;

[Figure]

Marlon et al., 2008).

Figure 7: The authors presented spatiotemporal patterns in the historical mean annual temperature anomalies from 1895 to 2010 as shown in Figure 6. Is it possible to organize or discuss the temperature variations over the last 3000 years separately for the east and west part of the study region, considering that the two parts of the study region represent very different temporal variations in temperature and it is possible that the same situation occurred in the past too. On the other hand, the Figure 6 represents the variations in mean annual temperature. It is recommended to present similar figures or results in the text because most of proxy records are sensi- tive to summer temperatures. The same suggestion is applied to moisture variations. Maybe a figure similar to the Figure 6 is helpful to guide the authors to organize the different archives based on the sub-regions in the study area. I guess that the moisture variations might have differed more regionally than temperature.

There are very limited hydroclimatic data available from the Midwestern part of our study area, but we can differentiate the western versus eastern bog records, and discuss the western tree-ring and bog data together before speaking about the remaining data, which effectively apply only to the eastern portion of the domain. Figure 8, which shows dominant patterns of spatial variations in drought from 1700-2005, is intended to be analogous to Figure 6, which shows the spatial variations in temperature in the region.

11) Figure 7: What do they mean by the '+' symbol? Please clarify it in the Figure legend. How about the dating uncertainties for lake or bog sediment records? It is crucial for comparison with other high-resolution data. It is possible that the dating accuracy affects poor correspondence among the proxies. An additional table outlining all the details including dating material, sampling resolution, temporal resolution, dating methods and so on is useful. As a result, a separate discussion paragraph is needed. What is the full name for NY, RI, ME and others? All the testate amoeba data are from the same region, west or eastern region? From Figure 1, the related data distribute

widely. Are they arranged regionally or not? Can they provide the simulation output results for moisture variations over the past 3000 years from GCM modeling? If so, it would be a very helpful addition.

The "+" symbol indicates time periods with high water table depths and thus signify drought events; we will add this to the figure caption. The dating uncertainties are always an important component of uncertainty in the sediment records, and they certainly have an impact on multi-proxy comparisons; we will provide a table containing details about the sediment records used in the analysis, with their sample and temporal resolution, dating methods, and related material. An additional paragraph discussing these differences in the proxy records can also be added. We will add the full names for the states, and also separate the testate amoeba records by region (they are currently organized by region but we can distinguish them in the figure as such). The moisture variations from the GCM modeling can also be included as requested. We greatly appreciate all of these helpful suggestions.

---

## Author Response (AR1)

**Paleoclimate Paper: Response to Reviewers**

Referee #1

Referee #1 requests two key areas in need of improvement in the paper: 1) revisions to Figures 5, 6, and 7 to facilitate understanding and interpretation of the paleoclimate data, and 2) additional information and explanation of the seasonality of different data sources and how this plays out in the reconstructions. We appreciate the recommendations and attention to detail and have corrected the manuscript as follows.

General corrections:
- Corrected axes, axis labels, panel labels, and legends for Figures 3, 4, 5, 6, and 7.
- Added explanations of the seasonality for each proxy in their descriptive sections, and noted these in the discussion of the features and trends in the multiproxy figures. The PDSI reconstruction is calculated for June, July, and August, which may help explain why it differs from the bog data.

Figure 4: This figure has 6 panels, adding the a, b, c, d, e, f for each panel is helpful for a smooth reading. Now the order of figure is confusing.

We added panel labels and changed the order of the panels to be more intuitive, moving from the most direct data comparisons (with the least uncertainty) to the more indirect comparisons (with more uncertainty).

Page15 line 9: Figure 5A, should be Figure 4e (New Long versus Deep Pond water elevations from southeastern MA for the past 7000 yrs).
Fixed.

Page15 line 12: Figure 5B should be Figure 4f (Deep Pond lake-level derived P-E reconstruction versus the mean annual precipitation reconstruction based on the mean of multiple records in the region).
Fixed.

Page15 line 19: Figure 5C should be Figure 4c (observed versus predicted paleohy-drological variables, based on proxies found in lake and bog archives, and inferred using space-for-time paleoclimatic transfer functions and validated using cross-validation. Left: pollen data from (Marsicek et al., 2013)).
Fixed.

Page15 line 31: Figure 5D should be Figure 4d (observed versus predicted paleohy-drological variables, based on proxies found in lake and bog archives, and inferred us- ing space-for-time paleoclimatic transfer functions and validated using cross-validation. right: testate amoebae data from across North America (Booth, 2008)).
Fixed.

Page16 line 4: Figure 5E should be Figure 4A (instrumental (observed) versus reconstructed values correlated in time. Left: NY PDSI for 1895-2000).
Fixed.

Page 16 line 23: Figure 1I should be Figure 1 or Figure 5I.
Fixed.

Page 16 line 24-26: The mean annual temperature anomalies based on pollen records from across North America from Viau et al. 2006 (Figure 5H) also show a long-term but more subtle decline.
Fixed.

Page 16 line 31-32: add more explanation on the reason why long-term difference between reconstruction by Williams et al. 2011 (Figure 5I) and Viau et al., 2006 (Figure 5H).

Added a paragraph on pg 17, line 6-15.

Figure 5: the x axis should not contain "0" for CE. Same for Figure 7.
Fixed.

Figure 6: add the loading legend. Which color indicates positive loading? Adding the a, b, c, d to each panel.
The sign of the loadings is arbitrary. The green on the map corresponds to the green temperature anomalies below, and likewise for the brown colors. Axis one scores are larger than Axis two scores, consistent with PCA more generally. We added labels to the panels.

Page 17 line 19-26. The sentence is not clear. Does brown line (Figure 6c) means temporal variations of EOF1?
The brown line refers to the negative loadings of EOF 1; we will clarify this sentence.

Page 18 line 12: during the past 9000 years should be 900 years. Figure 7c does not show clear long-term trend. Maybe it does not preserve low frequency signal so much.
Fixed.

Page 18 line 14-15: Could you give explanation why there is no correlation between tree-ring based PDSI and varve-based index? Both tree ring and varve records could be calibrated with instrumental data, but they are not correlated. Please give more information on reconstructed PDSI, seasonal PDSI? Or annual? Same as varve-based reconstruction.
The apparent lack of correlation calculated from the PDSI and varved record was due to differences in the spatial domain for the PDSI reconstruction; in fact the two series are well correlated, most strongly on a mean annual basis, but we edited this part of the text and removed the issue entirely.

Referee #2

Referee #2 requests a general revision of sections 3.3. and 3.4 (now sections 4.2 and 4.3) to improve the clarity and understanding of the data interpretation. We appreciate this feedback and agree that some restructuring of the text would be helpful. Currently, we describe the oldest features in the temperature and hydroclimate data, and move towards present, instead of walking through each of the proxies in some kind of priority order. We chose this approach because each of the proxies has different strengths and weaknesses, so in fact we would argue that there is not a natural "priority" as to which is best. In addition, a primary objective of the paper is to understand whether the proxies share common features and signals when considering them all together, which is why we use a chronological framework for interpreting the data. However, we appreciate that our approach is not clearly stated at the beginning of these sections, and thus may be confusing. To address this, we re-wrote these two sections for clarity based on the new ordering of the data in Figures 5 and 7. We also added an explanation at the top of section 4.2 explaining our reasoning. We also agree that grouping model data together will help reduce confusion in these sections, which we note in the specific comments below.

The sections 3.3 and 3.4 became 4.2 and 4.3 because we moved the Methods closer to the actual data analyses, and moved the discussion about the uncertainties directly after the proxy descriptions.

1) Page 3, line 29, I am confused with 'bogs with groundwater-sourced inputs'. Do you mean that the bog is influenced by groundwater?

Pg 3, line 27: Yes, we clarified to say simply groundwater.

2) Page 10, line 13, please rephrase this sentence.

Pg 10, line 21: Removed "Newby and colleagues".

3) Page 16, lines 18-21, it is important to clarify why the two reconstructions have very different trends considering both used the same dataset.

Agreed. We explain this also in our response to Referee #1, reproduced here:
The reconstructions by Viau et al. 2006 and Williams et al. 2011 reflect differences in the underlying fossil pollen data, age models, the modern calibration datasets, and the reconstruction method, although the methodological differences are likely the least important factor. Viau et al. 2006 used 752 pollen records and 4590 modern calibration samples. We used 863 pollen records from Williams et al. 2011, which had updated age models based on linear interpolation between age controls (Williams et al. 2004) and 4833 modern calibration samples (Whitmore et al. 2005). Both studies used slightly different versions of the modern analogue technique,

which numerically compares the differences in fossil pollen composition of each sediment sample with the composition of each modern sample from a calibration dataset. The environmental characteristics of the most similar modern samples are then averaged and assigned to the target fossil sample (Williams et al. 2008).

Explanatory paragraph added on pg 17, line 6-15.

4) Page 18, lines 11-12 are confusing and need clarification. It is not the case by visual inspection.

We appreciate this note – this is an error from an earlier iteration of the figure and the text has been updated to reflect the new figure on Pg. 19 starting at line 8.

5) Page 20, line 6: Visual inspection did not show that bog water levels were lower at that time.

This is also correct and we fixed this text now at Pg. 21 line 10.

6) Figure 1: 'Proxy type' should be proxy type and sites? It is unclear that each of the pollen sites hosts a long-term climate history?

Agreed and fixed. We are not certain of the second question here. There is a long history of using pollen records to reconstruct climate, which we describe in the proxy section.

7) Figure 2: What do you mean by "E" in the Figure 2? It needs full name. The same is for Q, and others. Please check on the figure 2 to be sure all the descriptions are correct and complete.

This was indeed missing – E stands for "Evaporation." This was fixed in Fig. 2 and its caption.

8) Figure 3: it is confusing and somewhat misleading why the temporal resolution (vertical axis) of trees varies from seasonal to 100 years. Please clarify and confirm it. The same question is for modern/Historical instruments.

We changed the y-axis label to say "Temporal climate variability reflected by data type" to clarify and clarified associated text on pg. 4, line 12.

9) In Figure 4, bottom row, right panel: correct 'precipition' as precipitation. What do you mean by 'NY, MN, and others'? Please provide the full name, and the compared common periods are also needed for clarification.

Fixed.

10) Figure 5: It would be more logical to arrange the simulation curves together. Anyway, the curves in Figure 5 need to be re-arranged. The pollen data F, H and I are very different from each other. Which is more reliable? The difference is caused due to regional temperature differences as identified in Figure 6 or different methods or dating uncertainties? More discussion is needed for clarification, rather than just listing them together. I don't know why the G and F figures were arranged reversely. Did the North central and northeastern US simulations correspond to the west and the Atlantic margin in Figure 6 respectively? Liu et al., 2010 did not match with Liu et al. 2009 in the reference list. The Liu et al. 2009 simulation represents a long-term decreasing trend which is similar to most of reconstructions, but it is different from reconstructions in term of on medium-frequency domains. Also, I miss some discussions on driving mechanism in the study region considering that they used the GCM simulation results. To my knowledge, Esper et al. (2012) identified an orbital forcing in tree-ring data in high NH latitudes (to the north of 65 degrees north latitude), but the forcing weakens quickly towards middle- and lower-latitudes. The study region in this manuscript is located in middle latitudes. Perhaps the orbital forcing is not strong or non-existing. Anyway, more discussion is helpful for mechanism hinted for the last 3000 years.

We agree that Figure 5 can be reorganized and explained in a clearer manner, grouping the model output in particular. The pollen data span different spatial scales, and for our domain the curve in panel I is the most reliable because it is specific to the northeastern U.S., and includes more records than the reconstruction in panel H (as explained above). This is noted in Pg. 17, line 14.

Regarding the north central and northeastern curves distinguished in the model data (panels G and J), yes, these do reflect areas similar to those highlighted by the EOF analysis in Figure 6. We explain this more clearly in the text now, and also correct the Liu et al. citation on the figure (it should be Liu et al. 2009). The mechanism behind the long-term decline in northern hemisphere temperatures observed in the pollen data and in the summer temperature simulations is orbital forcing, declining northern hemisphere summer insolation in particular (Liu et al., 2014). Clarified in Pg. 17 line 17.

Figure 7: The authors presented spatiotemporal patterns in the historical mean annual temperature anomalies from 1895 to 2010 as shown in Figure 6. Is it possible to organize or discuss the temperature variations over the last 3000 years separately for the east and west part of the study region, considering that the two parts of the study region represent very different temporal variations in temperature and it is possible that the same situation occurred in the past too. On the other hand, the Figure 6 represents the variations in mean annual temperature. It is recommended to present similar figures or results in the text because most of proxy records are sensi- tive to summer temperatures. The same suggestion is applied to moisture variations. Maybe a figure similar to the Figure 6 is helpful to guide the authors to organize the different archives based on the sub-regions in the

study area. I guess that the moisture variations might have differed more regionally than temperature.

There are very limited hydroclimatic data available from the Midwestern part of our study area, but we can differentiate the western versus eastern bog records, and discuss the western tree-ring and bog data together before speaking about the remaining data, which effectively apply only to the eastern portion of the domain. Figure 8, which shows dominant patterns of spatial variations in drought from 1700-2005, is intended to be analogous to Figure 6, which shows the spatial variations in temperature in the region. The bog data are now displayed differently (separating east and west sites) in Fig. 7.

11) Figure 7: What do they mean by the '+' symbol? Please clarify it in the Figure legend. How about the dating uncertainties for lake or bog sediment records? It is crucial for comparison with other high-resolution data. It is possible that the dating accuracy affects poor correspondence among the proxies. An additional table outlining all the details including dating material, sampling resolution, temporal resolution, dating methods and so on is useful. As a result, a separate discussion paragraph is needed. What is the full name for NY, RI, ME and others? All the testate amoeba data are from the same region, west or eastern region? From Figure 1, the related data distribute widely. Are they arranged regionally or not? Can they provide the simulation output results for moisture variations over the past 3000 years from GCM modeling? If so, it would be a very helpful addition.

The "+" symbol indicates time periods with high water table depths and thus signify drought events; we will add this to the figure caption. The dating uncertainties are always an important component of uncertainty in the sediment records, and they certainly have an impact on multi-proxy comparisons; we provide "Table 1" now containing details about the sediment records used in the analysis, with their resolution and related material. It is not practical or even possible, however, to obtain and re-assess all the dates in the original publications in this synthesis. We did add full names for the states, and also separate the testate amoeba records by region (Fig. 7). The moisture variations from the GCM modeling are also now included (Fig. 7). We greatly appreciate all of these helpful suggestions.

[revised manuscript text omitted]